# HOW2EVERYTHING: Mining the Web for How-To Procedures to Evaluate and Improve LLMs

Yapei Chang [1 2]   Kyle Lo [2 3]   Mohit Iyyer [1]   Luca Soldaini [2]

## Abstract

Generating step-by-step "how-to" procedures is a key LLM capability: how-to advice is commonly requested in chatbots, and step-by-step planning is critical for complex reasoning tasks. Yet, measuring and improving procedural validity at scale on real-world tasks remains challenging and understudied. We introduce HOW2EVERYTHING,[2] a scalable framework to evaluate and improve goal-conditioned procedure generation. Our pipeline HOW2MINE extracts and rewrites 351K procedures from 980K web pages across 14 topics, and can scale to larger corpora. From this pool we build HOW2BENCH, a 7K-example evaluation set balanced across topics. We also introduce HOW2SCORE, an evaluation protocol that uses an LLM judge to detect whether a generation contains any *critical failure* that would prevent achieving the goal. For low-cost, reproducible evaluation, we distill a frontier judge into an open 8B model achieving 80.5% agreement with annotators. HOW2BENCH shows clear scaling trends across model size and training stages, providing signal early in pretraining. Finally, RL with HOW2SCORE as a reward improves performance on HOW2BENCH by >10 points across three models without systematic regressions on standard benchmarks, with gains robust to superficial source-document memorization or format compliance. We release all code and data.

## 1. Introduction

The ability to understand and generate how-to procedures is a key capability for large language models (LLMs). On one

[1]University of Maryland, College Park [2]Allen Institute for AI [3]University of Washington. Correspondence to: Yapei Chang <yapeic@umd.edu>.

*Proceedings of the 43rd International Conference on Machine Learning*, Seoul, South Korea. PMLR 306, 2026. Copyright 2026 by the author(s).

### The HOW2EVERYTHING Framework

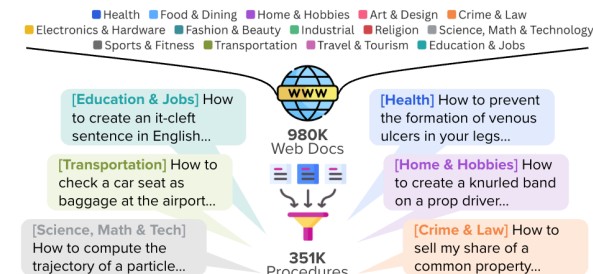

*Figure 1a* **HOW2MINE** extracts and refines diverse how-to procedures from the web. Using this pipeline, we extract 351K samples from nearly 1M web pages across 14 topics.

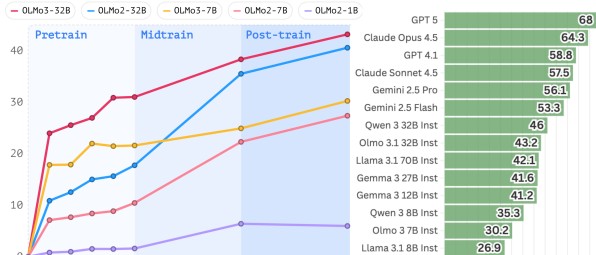

*Figure 1b* We reserve 7K samples for **HOW2BENCH**. This benchmark measures performance for a broad range of models (from 1B parameters to frontier LLMs), and stages of development (pretrain to post-train), making it ideal for scaling laws. The benchmark uses **HOW2SCORE**, a cost-effective, open-source LLM judge model that achieves strong agreement (80.5%) with human annotators.

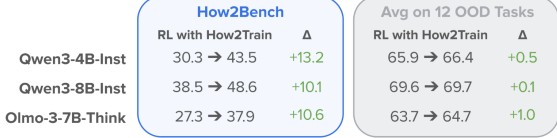

*Figure 1c* Across instruct and thinking models, **RL on our web-mined procedures** alone lifts performance on HOW2BENCH while maintaining or improving scores on a standard post-train evaluation suite (Olmo et al., 2025), suggesting that procedure generation is a capability that is broadly useful across downstream tasks.

hand, LLMs are increasingly used for practical how-to guidance: approximately 8.5% of ChatGPT conversations fall under the *How-To Advice* category (Chatterji et al., 2025), with open-access conversation datasets like WildChat and

[2]Of course, no method has infinite coverage. The name is a playful pun to convey the scale and diversity of our framework.

*Table 1.* Given a goal, list of resources, and target number of steps $N$ (inferred from reference doc), the model generates $N$ steps. HOW2SCORE evaluates the generation by checking whether it contains any critical failure using our distilled LLM judge (§4.4). For each example, we show a brief description of a representative critical failure, highlighting relevant steps. More examples in §G.1.

| Example (goal + resources + reference steps) | Model-generated steps | Critical failure |
|---|---|---|
| **Goal:** [Crime & Law] Sell your share of a common property apartment by following the required legal procedure for notifying co-owners and transferring ownership.

**Resources:** notary; notarial document; letter with a list of contents; receipt.

**Reference: 1.** Prepare a notification to all co-owners stating the conditions of sale of your share. **2.** Visit a notary to draw up a notarial document including all sale conditions. **3.** Distribute the notarial document to all co-owners by letter with a list of contents, obtaining a receipt from each. **4.** *Wait 30 days* for co-owners to express their desire to purchase your share. **5.** Sell your share to a third party. | **Generator:** Gemini 2.5 Pro

**1.** Engage a notary to draft a formal notification of the intended sale for all co-owners. **2.** Send the notification via a letter with a list of contents and keep the postal receipt as proof. **3.** Instruct the notary to prepare the final sale agreement as a notarial document. **4.** Execute the transfer by signing the notarial document in the presence of the notary. **5.** Receive the registered notarial document and the final transaction receipt from the notary. | **Skips** a required prerequisite (the 30-day waiting period) before completing the transfer. |
| **Goal:** [Food & Dining] Prepare spiced glazed nuts with a crisp candy glaze using a Crock Pot.

**Resources:** Crock Pot; nuts; melted butter; powdered sugar; spices; racks; waxed paper.

**Reference: 1.** Preheat the Crock Pot on high for 15–20 minutes. **2.** Add nuts to the Crock Pot. **3.** Pour melted butter over the nuts and stir. **4.** Add powdered sugar and stir until completely blended. **5.** Cover and cook on high for 15 minutes. *6. Remove the lid, set the Crock Pot to low, and cook with the lid off for 2–3 hours, stirring occasionally* until the nuts have a crisp glaze. **7.** Stir spices over the nuts and stir to distribute. **8.** Pour nuts onto racks or waxed paper and let cool completely before serving or bagging. | **Generator:** Qwen 3 8B Instruct

**1.** Place the nuts in the Crock Pot. **2.** Add melted butter, powdered sugar, and spices to the Crock Pot. **3.** Cook on low heat, stirring occasionally, until the glaze thickens. **4.** Transfer the nuts to racks to cool and dry. **5.** Cover the cooled nuts with waxed paper to maintain crispness. *6. Let the glaze set for several hours.* **7.** Ensure the nuts are completely dry and crisp. **8.** Serve or use as desired. | Leaves critical cooking times and lid/heat settings **unspecified** (e.g., high vs. low and lid on vs. off), making the procedure hard to execute reliably. |

LMSYS-Chat (Zhao et al., 2024; Zheng et al., 2024) showing similar trends (see Appendix §A). On the other hand, exposure to procedural content at all stages of model training has been shown to improve downstream tasks that rely on reasoning and planning (e.g., pretraining: Ruis et al., 2025; midtraining: Zhang et al., 2020; post-training: Brahman et al., 2024). Therefore, the ability to evaluate and improve this core skill across a diverse set of procedures has the potential to unlock progress on many downstream tasks.

Evaluating and optimizing end-to-end procedural validity is challenging in open-world settings, where real-world procedures span diverse goals and domains without a task-specific executor or automatic oracle. One key challenge is *diversity*: existing work is limited to narrow domains like cooking recipes (Lal et al., 2024; Toyooka et al., 2025) or specific sources like how-to sites (Zhang et al., 2020; Yuan et al., 2023). Furthermore, successful procedure generation hinges on the validity of an entire sequence of actions, requiring *end-to-end evaluation*: prior work often focuses on subtasks, such as graph edge prediction (Sakaguchi et al., 2021) or step ordering (Zhang et al., 2020; Lal et al., 2024; Anika & Miah, 2025). Finally, large-scale investigation requires *accurate yet efficient metrics*: string-overlap metrics like BLEU are fast to compute but inaccurate, and human annotation is accurate but expensive (Brahman et al., 2024).

To fill these gaps, we introduce **HOW2EVERYTHING**, a scalable framework to evaluate and train models for step-by-step procedure generation. It combines HOW2MINE, a web-scale pipeline to mine and refine procedures, HOW2BENCH, a benchmark to assess performance, and HOW2SCORE, an efficient LLM-as-a-judge model to detect critical failures in LLM-generated steps. Our contributions are:

**Contribution 1: A pipeline to collect realistic, diverse procedures at web-scale.** Rather than drawing from narrowly scoped how-to websites, **HOW2MINE** can scale to arbitrarily large collections of web documents (Figure 1a). It ensures broad coverage by sampling from 14 different topics, as identified by WebOrganizer (Wettig et al., 2025). Using multiple stages of filtering and refining, we remove low-quality procedures and standardize format. We show the effectiveness of this pipeline by processing 980K web documents to derive 351K procedures (§3).

**Contribution 2: An accurate, low-cost, and reproducible protocol to evaluate procedure generation.** All existing protocols have limitations: efficient automatic metrics (e.g., perplexity on reference procedure, string overlap between reference and model generation) are unreliable (Lyu et al., 2021; Li et al., 2023), human annotations are expensive and slow, and solely relying on LLM APIs as a judge is not reproducible. Therefore, we establish **HOW2SCORE**, an evaluation protocol that checks whether a generated procedure contains a *critical failure*, meaning an omission, an extraneous action, or a deviation that would prevent achieving the goal under the stated constraints (see Table 1 for examples). We first validate this protocol by assessing agreement between frontier LLM APIs as judges and human annotators on 200 examples, and note that, across 5 LLMs, agreement with human majority is consistently high, ranging from 76.5 to 83.0%. Then, we use labels from LLM APIs to distill a compact, 8B model for repeatable and efficient evaluation. This model maintains high agreement with annotators (80.5%), enabling efficient assessment (§4).

**Contribution 3: Mined procedures benchmark performance across large range of model sizes and capabilities.** We reserve 7K of the 351K mined procedures for evaluation

(500 instances per topic) and create **HOW2BENCH**. We find that HOW2BENCH can meaningfully rank models trained on vastly different amounts of compute (from just $\approx 10^{21}$ FLOPs—e.g., a 1B model trained on 200B tokens—to the latest frontier models, such as GPT 5 (OpenAI, 2026)), and also compare base models against instruct variants. This is a desirable property for a benchmark, as it enables ranking models (Heineman et al., 2025) and establishing scaling laws (Xu et al., 2025) across compute budgets; however, many benchmarks either saturate early in training or target frontier models, having near-zero performance at smaller scales (e.g., Kazemi et al., 2025). In contrast, across training runs spanning 1B–32B models, HOW2BENCH shows clear scaling trends with both model size and training stage (Figure 1b; Table 8), enabling the study of techniques to improve this key skill across the entire model training pipeline (§5).

**Contribution 4: Training on procedures improves how-to generation, with robust gains.** Beyond measuring performance, the same scalable components that make HOW2EVERYTHING cheap to evaluate also make it practical to optimize models on this task (Figure 1c). HOW2SCORE provides an actionable reward signal for RL. Across three models, RL with HOW2SCORE as a reward consistently improves HOW2EVERYTHING performance by >10 points. Importantly, these improvements do not introduce regressions—and, in some cases even improve—on a standard post-training evaluation suite (Olmo et al., 2025), suggesting that procedure generation is a capability useful across tasks (§6). We further show that these gains are not driven by superficial format compliance or source-document memorization: improvements in base models consistently yield better checkpoints after RL, and even after aggressively contaminating data, performance on procedure generation only improves by a modest amount (+3 points for a 7B model; §7).

## 2. Problem Setting and Related Work

In this work, we use "procedure" to refer to a goal-conditioned sequence of actions. We distinguish between *descriptive procedures*, where a model can generate textual representations of such sequences, and *executable procedures*, where correctness is determined by execution—either in grounded environments with explicit state transitions, such as formal transition systems (Samiei et al., 2025) or simulated environments (Puig et al., 2018; Shridhar et al., 2021), or through internally executed reasoning strategies for problem solving (Mao et al., 2024; Ruis et al., 2025). Our work focuses on real-world procedures, which fall under the first category. While a model cannot actually file for divorce, the ability to accurately represent the steps involved remains a core user-facing capability and a necessary

prerequisite for downstream systems that aim to support or automate parts of real-world processes.

**Datasets for goal-conditioned procedures.** Within the descriptive procedural setting, dataset construction has typically been constrained along two axes, limiting coverage of diverse real-world procedures. Many datasets are restricted by topical domain, with common choices including cooking Bień et al., 2020; Toyooka et al., 2025; Anika & Miah, 2025. Others are restricted by collection source, such as instructional platforms like WikiHow and Instructables (Zhou et al., 2022; Bolotova-Baranova et al., 2023; Brahman et al., 2024; Uzunoglu et al., 2024). Our work goes beyond these constraints by mining naturally occurring, goal-conditioned procedures across 14 topics from the web. Adjacent work studies expert-specified methodical writing tasks with concrete instantiations across domains (Malaviya et al., 2024).

**Evaluation challenges for end-to-end procedural validity.** Given these datasets, prior work has adopted a range of task formulations to study procedural capabilities. Examples include edge prediction over step pairs (Sakaguchi et al., 2021), step reordering (Anika & Miah, 2025), QA (Lal et al., 2024; Uzunoglu et al., 2024), or constraint satisfaction (Yuan et al., 2023). These formulations test local causal-temporal relations, implicit planning knowledge, or reference similarity, whereas HOW2BENCH evaluates complete generated procedures for end-to-end validity via critical-failure detection. In the setup where the task is pure generation (closest to ours), models directly generate a step sequence given a goal and are typically evaluated with perplexity or string-overlap metrics such as BLEU (Lyu et al., 2021; Li et al., 2023; Sakaguchi et al., 2021; Brahman et al., 2024). Prior work, however, acknowledges that these metrics are insufficient proxies for procedural validity and relies on human evaluation for more reliable signal (Lyu et al., 2021; Brahman et al., 2024). More recently, LLM-as-a-judge protocols have been used as a general approach to scale evaluation for open-ended generation (Zheng et al., 2023; Dubois et al., 2025), but generic preference-style judging can overemphasize surface qualities like coherence or helpfulness, and may fail to predict downstream plan usefulness (Balepur et al., 2025). Closest to our use of web evidence for evaluation, Wadhwa et al. (2025) retrieve instructional documents to synthesize task-specific evaluation criteria. Together, these approaches highlight a fundamental reliability–scalability tradeoff, which we address by introducing a validity-oriented evaluation protocol.

## 3. HOW2MINE: Extracting Realistic Step-by-Step Procedures from the Web

To evaluate end-to-end procedural validity at scale, we mine goal-conditioned step-by-step procedures from a large web corpus to ensure broad topical coverage. As a proof of con-

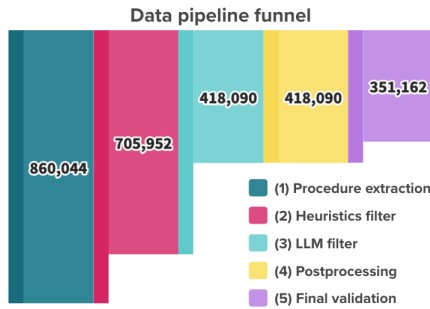

*Figure 2.* Given a sample of 980K topic-stratified web documents, HOW2MINE yields 351K *procedure instances* (goal + resources list + reference steps), and can be easily scaled to larger corpora.

cept, we run HOW2MINE on 980,000 web documents to extract 351,162 structured procedure instances (Figure 2). The pipeline scales straightforwardly to larger corpora, making it possible to dynamically construct evaluation sets and training corpora without manual curation.

### 3.1. Sampling Web Pages for Procedure Mining

We source candidate documents from the DCLM web corpus (Li et al., 2025). Because tutorial-style documents tend to have a high density of explicitly ordered, imperative steps, we restrict our document pool to those labeled as *Tutorial & How-to Guide*[3] by the WebOrganizer format classifier (Wettig et al., 2025). To ensure equal topical coverage, we apply the WebOrganizer topic classifier and perform stratified sampling across 14 topics.[4] Our final pool of 351K procedure instances spans 189K unique domains (we report top 10 domains per topic in §B.1).

### 3.2. From Web Documents to Structured Procedures

Starting from this topic-stratified pool of tutorial documents, we run a multi-stage pipeline to extract, filter, and post-process procedures. All LLM-based stages use GPT-4.1 (OpenAI, 2025), see prompts in §H.1. Using the OpenAI batch API, running this pipeline over 980K documents issues 252K requests, costing 5,717 USD.

**(1) Procedure extraction.** Given a candidate web document, we first use an LLM to identify whether it contains a well-formed sequential procedure and, if so, extract the goal and an ordered list of steps.

**(2) Heuristics filter.** We run simple heuristics-based checks to remove (i) candidates with fewer than 5 or more than 15 steps to avoid trivial or overly complex procedures, and (ii) those with high n-gram overlap within the extracted steps. See §B.3 for implementation details.

---
[3]HOW2MINE can be easily extended to extract valid procedures from other formats such as academic writing and knowledge articles (see §B.2); for simplicity, we focus on a single format.

[4]See `weborganizer.allen.ai` for definitions and examples.

**(3) LLM filter.** We apply an LLM-based filter to exclude examples that (i) depend on specific named entities, (ii) are purely mathematical calculations, (iii) require interacting with UI elements, (iv) involve open-ended creative generation, (v) are non-sequential, or (vi) are unreasonable/nonsensical. These criteria are derived from multiple rounds of data inspection (see §B.4 for the in-depth rationale).

**(4) Post-processing and resource extraction.** For each remaining example, we rewrite the goal to be as specific and deterministic as possible, explicitly stating the required constraints and expected outcome. Because multiple distinct procedures can still satisfy a goal, we additionally list the resources (if any) referenced by the steps in the reference procedure. See Table 1 for examples. Together, these edits narrow the space of valid solutions.

**(5) Final validation.** Finally, we run an LLM-based sanity check to remove any remaining nonsensical or otherwise invalid procedures.

**Pipeline outputs.** Each procedure instance is a structured record that includes a topic, goal, list of resources (possibly empty), and reference steps. §B.5 shows one full example for each topic. From this pool, we construct HOW2BENCH by sampling 500 instances per topic (7,000 total), and reserve the rest as training data.

## 4. HOW2SCORE: Measuring Procedural Validity by Detecting Critical Failures

Evaluating procedural generation comes with a trade-off between *scalability* and *reliability*. Reference-overlap metrics are cheap but miscalibrated to procedural validity, while human evaluation is reliable but does not scale (Li et al., 2023; Brahman et al., 2024). We introduce HOW2SCORE, a validity-based metric that asks whether a generated procedure contains any *critical failure* that prevents achieving its goal; the metric has high agreement with human annotators, and is easy to distill into an efficient LLM judge.

### 4.1. Defining Critical Failures in an Open-World Setting

We take inspiration from the framing commonly used in process reward models (PRMs) for mathematical reasoning (Lightman et al., 2023), where the earliest incorrect step identified by a verifier is treated as the point of failure. In open-world procedures, however, steps are not directly executable, making it difficult to localize a "first failure" automatically. We therefore develop a working definition and codebook by qualitatively inspecting model outputs and iterating with human annotators.

**Definition.** We define a *critical failure* as an omission, extraneous action, contradiction, severe vagueness, or other deviation from the reference that is severe enough to prevent

achieving the goal, or to make the procedure unusable as instructions. We use the reference procedure as an anchor, but aim not to penalize alternative valid procedures or superficial differences. For example, if the goal is to make a terracotta pot as a gift, a different gift message than the reference is not a critical failure. While what constitutes critical is inherently subjective in this non-executable setting, this definition provides a practical proxy. Table 1 shows representative *critical* failures; §G.1 provides additional examples, including non-critical variations.

**Assumption of reference correctness.** While rigorous filtering (§3) reduces noise, some references can still contain errors, and HOW2SCORE may inherit this noise. As a sanity check, we prompt GPT-4.1 to judge if each HOW2BENCH reference procedure reasonably achieves the stated goal; it accepts 96.6% of examples as valid. We additionally manually inspect 200 randomly sampled source pages: 191/200 come from credible sources, extracted procedures are reasonable in 167/200 cases, unclear in 28/200 cases requiring domain expertise, and unreasonable in 5/200 cases. In our formulation, we use $S^\star$ to make the task more deterministic and suitable for evaluation, not as a perfect ground-truth solution.

### 4.2. Evaluation Protocol

Given an evaluation set $D$ of examples $x = (g, R, S^\star, \hat{S})$ (goal $g$, extracted resource list $R$, reference procedure $S^\star$, and model-generated procedure $\hat{S}$), we use an LLM judge to identify *critical failures*. Each failure is accompanied by a description and references to the relevant steps in $S^\star$ and/or $\hat{S}$. We provide the full annotation codebook and examples of non-critical vs critical cases in the judge prompt in §H.3.

**Binary score aggregation.** From the judge output list, we derive a binary label: we assign `has_failure` if at least one critical failure is identified, and `no_failure` otherwise. To report performance over $D$, we aggregate the binary labels into a success rate (the fraction of examples labeled `no_failure`). Formally,

$$\text{Score}(D) = \tfrac{1}{|D|} \sum_{x \in D} \mathbb{I}\big[J(g, R, S^\star, \hat{S}) = \texttt{no\_failure}\big].$$

where $J(\cdot)$ denotes the derived binary judgment, answering the question: "Does this procedure contain *any* critical failure?" Compared to checking for the first failure as in the math PRM setup, this aggregation yields higher inter-annotator agreement (see §4.3). We therefore adopt this formulation, which remains aligned with our downstream objective. For transparency, we still ask the judge to enumerate all identified failures.

### 4.3. Validation via Human Annotations

To validate our definition of critical failures, we ask human annotators to list all critical failures they observe using the evaluation protocol in §4.2. We recruit three annotators via Prolific to label 200 examples (pre-screened to avoid procedures requiring specialized domain knowledge), paying an average hourly rate of 28 USD (total cost: 3,600 USD). See §C.1 for details.

**Annotator training and pilot studies.** In early pilots (300 annotations), many annotators either flagged *any* difference from the reference as critical, or overlooked indisputable failures masked by coherent surface form.[5] As a result, initial inter-annotator agreement was low (Krippendorff's $\alpha = 0.273$). We iteratively refined the training materials and added more examples to clarify the boundary between non-critical variations and critical failures. For the final round, we screened annotators with a short qualification test and selected the three who best demonstrated understanding to label 200 examples.

**Inter-annotator agreement.** With binary score aggregation, we observe Krippendorff's $\alpha = \mathbf{0.593}$. Given the non-executable, open-world setting and the existence of multiple valid procedures per goal, we do not expect near-perfect agreement; instead, we target a metric that is stable for relative comparisons (§5) and usable as an RL reward (§6). If we instead require agreement on the *location* of the first failure (as in math PRMs), agreement drops ($\alpha = 0.307$), motivating our use of binary aggregation.

**Evaluating LLM judges against human labels.** We obtain annotations from various LLM judges on the same 200 examples used to obtain human annotations, and compute their percentage agreement with the human majority labels. As shown in Figure 3, GPT 5 has the highest overall agreement (83.0%) and is well-calibrated across classes (83.7% on human-majority `has_failure` cases; 82.4% on `no_failure` cases). To contextualize these results, we measure leave-one-out agreement among human annotators, which ranges from 84.7% to 88.5%. GPT 5's agreement falls within a few percentage points of this range, suggesting performance comparable to individual annotators.

### 4.4. Distilling a Cost-Effective Judge

While GPT 5 (OpenAI, 2026) is shown to have strong agreement with human labels in §4.3, evaluating 7,000 examples with it costs around $15. We therefore use GPT 5 as a teacher judge and distill it into a smaller Qwen 3 8B model for stable, low-cost large-scale evaluation. We collect 73K

---

[5]One example of such a failure is when the procedure first says "cut the wood board into 5 pieces of equal size", but later says to "place the pieces on the table with the largest piece on top and the smallest piece on the bottom". This is a clear inconsistency.

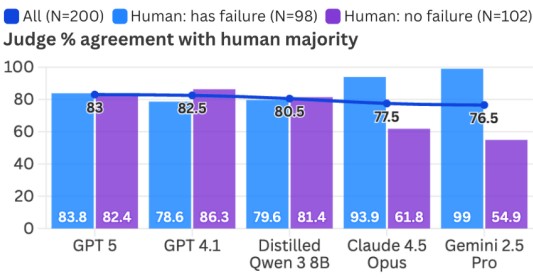

*Figure 3.* Agreement between LLM judges and the human majority label on critical-failure detection (N=200), reported overall and stratified by the human-majority class (`has_failure`/`no_failure`). §H.3 for the judge prompt; §C.1 for annotation details.

GPT 5 annotations on outputs from a diverse set of generator models,[6] and deduplicate to remove any overlap with the human-annotated set in §4.3. We then finetune Qwen 3 8B on this dataset for three epochs. On the human-labeled set in §4.3, the distilled judge achieves 90.5% agreement with GPT 5 and **80.5%** agreement with the human majority label. It is also relatively well-balanced across classes (79.6% agreement on `has_failure`; 81.4% on `no_failure`), making it a cost-effective alternative for large-scale evaluation (§5) and for serving as a reward function for RL training (§6). See more details on distillation in §C.2.

# 5. HOW2BENCH: Evaluating Performance on Step-by-Step Procedure Generation

Equipped with a reliable and scalable evaluation protocol, we run systematic evaluations on a range of models. We create HOW2BENCH by sampling 500 procedures per topic from the data created in §3, totaling 7,000 examples.

## 5.1. Inference Setup

At inference time, the model receives the goal $g$, resource list $R$, and required step count $n = |S^\star|$, and is asked to output a procedure $\hat{S}$ with exactly $n$ steps. While conditioning generations on $R$ and $n$ may not reflect real-world usage, it is an evaluation control to reduce degrees of freedom and improve comparability across model outputs. We enforce length control by requiring each step to be a single, concise sentence containing one main action, and asking the model to closely follow the concision level in the provided examples. See setup details in §D.1 and prompts in §H.2.

## 5.2. Evaluation Results and Analysis

We use HOW2SCORE with our distilled judge (§4.4) to evaluate a range of open and closed models, and report the main results in Figure 1b (more results in §D.3). Performance scales with model size and training stage, and we observe a

noticeable gap between open and closed models.

**No evidence of LLM judge self-preference bias.** A common concern with LLM-as-a-judge evaluation is self-preference: a judge might favor outputs from models in its own family (Zheng et al., 2023). We collect outputs from models in the GPT, Gemini, and Claude families and re-compute HOW2SCORE using models from these families as judges. Although absolute values vary, the relative model ranking is unchanged. See §D.2 for details.

**HOW2BENCH shows clear scaling behavior across model sizes and training stages.** Besides comparing fully trained *base* and *instruct* models, we also measure performance at intermediate steps of model training pipeline using pretraining, midtraining, and post-training Olmo checkpoints (OLMo et al., 2025; Olmo et al., 2025). 1b shows that across five Olmo training runs (Olmo 2: 1B/7B/32B; Olmo 3: 7B/32B),[7] HOW2BENCH exhibits smooth scaling across both model size and training stage, with a consistent ordering of model performance. We also observe the emergence of non-trivial performance by about 5% into pretraining for a 1B model (around $10^{21}$ training FLOPs), after which performance continues to improve. The sensitivity of HOW2BENCH to gains from all training indicates that it probes a core, general capability, making it well-suited for performance forecasting (Xu et al., 2025).

**Early emergence of procedural formatting.** We find that surface-level procedural formatting stabilizes early in training, particularly for larger models. We track simple formatting proxies across checkpoints: step-count mismatch relative to the reference, duplicate-step frequency, and $n$-gram repetition. Over five OLMO runs, formatting errors drop during early pretraining and quickly plateau while HOW2SCORE continues to improve. This decoupling resembles an emergence-like pattern: surface-formatting behavior stabilizes early, while procedural validity keeps improving. Thus, the continued gains we observe later in training are unlikely to be driven primarily by correcting surface-formatting errors, and instead reflect improvements in end-to-end procedural validity (details in §D.5).

**HOW2SCORE is not simply reducible to reference-step likelihood.** To test whether our task is simply reducible to perplexity over the reference, we compare checkpoint ordering under HOW2EVERYTHING to checkpoint ordering by conditional perplexity on the reference steps. Across five Olmo runs,[8] the Spearman correlation between checkpoint rank by HOW2EVERYTHING and rank by perplexity ranges from 0.233 (Olmo 2 32B) to 0.967 (Olmo 2 1B),[9] indicat-

---

[6]These include three 1B checkpoints, four 7B checkpoints, three 32B checkpoints, and four closed-source models.

[7]Olmo checkpoints corresponding to midtraining stage are labeled "stage 2 pretraining"; see Table 8 for exact IDs.

[8]We use 9 checkpoints per run: 8 stage-1 pretraining checkpoints plus the stage-2 midtrained checkpoint.

[9]With intermediate values 0.667 (Olmo 2 7B), 0.867 (Olmo 3

ing that HOW2SCORE is not simply measuring conditional likelihood of the reference procedure (full results in §D.6).

**Controlling for topic, required step count is a monotonic difficulty knob.** To better interpret aggregate scores and enable difficulty-controlled slices of HOW2BENCH, we examine simple instance properties that correlate with HOW2SCORE. We find that reference step count $|S^\star|$ is the dominant predictor across models: procedures requiring more steps are consistently harder, making $|S^\star|$ a simple, monotonic difficulty knob (details in §D.7).

**Qualitative examples of common failure patterns.** To study which non-formatting failures occur, we perform a small-scale qualitative analysis over model generations. While we occasionally observe refusals (primarily in frontier models), most errors fall into the following types: critical omissions of required actions; missing parameters (e.g., times, quantities, temperatures) that make steps non-executable; wrong values for critical parameters; unsafe or invalid actions; and internal contradictions across steps. See §G.1 for more details on this analysis.

## 6. RL on Step-by-Step Procedure Generation

Beyond serving as an evaluation protocol, HOW2SCORE can also be used as a practical RL reward for improving goal-conditioned step-by-step procedure generation, with gains that persist under external judges and without systematic regressions on standard out-of-domain benchmarks. These results suggest HOW2EVERYTHING provides a practical framework for both evaluating and improving goal-conditioned step-by-step procedure generation, and that HOW2SCORE-based RL can complement existing post-training pipelines as an additional optimization target.

### 6.1. Training Setup

We create a training set by sampling 100K examples from HOW2MINE, balanced across 14 topics and with low semantic similarity to HOW2BENCH instances (see §E.1).

For SFT, we fine-tune base and instruction-tuned checkpoints of Qwen 3 4B and 8B (Qwen Team, 2025a), and OLMo 3 7B (Olmo et al., 2025) for one epoch. For RL, we train Qwen 3 4B Instruct and 8B Instruct (Qwen Team, 2025a), and OLMo 3 7B Think (Olmo et al., 2025),[10] using Group Relative Policy Optimization (GRPO) (Shao et al., 2024) for 1000 optimizer steps with three rewards: (i) HOW2SCORE computed by our distilled judge, (ii) a step-format verifier, and (iii) a reference-calibrated length

reward to prevent length gaming. See §E.2 for full details.

### 6.2. Results and Analysis

**Length control effectively prevents length gaming in RL.** With the reference-calibrated length reward, generations stay close to the reference length ($|\text{gen}|/|\text{ref}| \approx 1.0$). Without it, models inflate length (up to $1.34\times$–$1.53\times$ the reference) and achieve large apparent HOW2BENCH gains consistent with length gaming (details in §E.4). This controls a major confound in LLM-as-judge settings where judges are prone to verbosity bias (§5.1).

**RL improvements persist under external judges.** To test whether these gains are specific to our distilled judge (which was used to compute HOW2SCORE during training), we re-evaluate the same RL-trained model generations with external judges (GPT 5 and Gemini 2.5 Pro), and find the gains persist. See §E.5 for details.

**RL improvements are reflected in blinded manual evaluation.** As a sanity check, we run a blinded manual evaluation on 200 randomly sampled HOW2BENCH examples comparing Qwen3-8B before vs. after RL. For each example, an author evaluator independently labels two anonymized outputs for critical failures, with the pre-/post-RL mapping randomized and hidden. The post-RL model improves from 32.5% to 40.0% no-failure, with 23 RL-only wins versus 8 base-only wins (McNemar exact $p = 0.0107$).

**RL improves HOW2BENCH performance without systematic out-of-domain degradation.** Table 2 shows that RL-trained models improve HOW2BENCH while retaining performance on standard out-of-domain evaluations: changes are mixed but generally modest (improving some benchmarks while regressing on others), with no evidence of systematic degradation. This suggests that our HOW2SCORE-based reward can complement existing post-training pipelines as an additional RL signal.

**Additional SFT stage yields limited gains on instruct checkpoints.** We find that SFT can yield small gains when applied to base model checkpoints, but does not improve instruction-tuned checkpoints (see §E.3). One plausible explanation is objective mismatch: SFT maximizes likelihood of a single reference text per goal, which need not align with minimizing critical failures under HOW2SCORE (Stiennon et al., 2022; Xie et al., 2025).

## 7. Is HOW2BENCH a Robust Evaluation?

Because HOW2BENCH is scored with an LLM judge and our evaluation examples are mined from the web, improvements in HOW2SCORE could plausibly arise from two confounding factors: (1) better compliance with an implicit task format, and (2) source-document memorization. We run

---

7B), and 0.483 (Olmo 3 32B).

[10]*Thinking mode* refers to the presence of explicit intermediate reasoning. Qwen models integrate instruction-following and reasoning in a single checkpoint, whereas OLMo provides separate Instruct and Think checkpoints.

*Table 2.* Results before and after RL with HOW2SCORE as reward (step 1000). We report performance on HOW2BENCH and 12 standard out-of-domain benchmarks. *Delta* rows show changes relative to the original checkpoint, and the final column $\overline{\Delta}_{\text{OOD}}$ is the mean out-of-domain change. In *Delta* rows, blue indicates positive change and orange indicates negative change.

| Model | In-domain | Out-of-domain | | | | | | | | | | | | |
|---|---|---|---|---|---|---|---|---|---|---|---|---|---|---|
| | HOW2BENCH | MMLU-Pro | GPQA | ZebraLogic | AlpacaEval | HumanEval+ | LiveCodeBench | MBPP+ | GSM8K | Minerva | Omega | AIME24 | AIME25 | $\overline{\Delta}_{\text{OOD}}$ |
| Qwen3-4B-Inst | 30.29 | 60.16 | 44.87 | 82.4 | 44.78 | 71.95 | 85.6 | 67.46 | 94.09 | 90.38 | 42.2 | 60.42 | 46.04 | - |
| + RL on HOW2TRAIN | 43.52 | 61.70 | 44.64 | 81.2 | 47.73 | 75.43 | 85.38 | 66.98 | 93.78 | 90.45 | 39.4 | 60.42 | 49.48 | - |
| *Delta* | *+13.23* | *+1.54* | *-0.23* | *-1.2* | *+2.95* | *+3.48* | *-0.22* | *-0.48* | *-0.31* | *+0.07* | *-2.8* | *0.00* | *+3.44* | *+0.52* |
| Qwen3-8B-Inst | 38.52 | 62.16 | 54.02 | 85.2 | 58.44 | 81.28 | 86.32 | 68.65 | 95.68 | 91.20 | 44.4 | 61.15 | 47.29 | - |
| + RL on HOW2TRAIN | 48.62 | 63.11 | 53.79 | 85.7 | 58.76 | 79.57 | 86.11 | 69.31 | 95.30 | 91.92 | 44.4 | 59.06 | 49.48 | - |
| *Delta* | *+10.10* | *+0.95* | *-0.23* | *+0.5* | *+0.32* | *-1.71* | *-0.21* | *+0.66* | *-0.38* | *+0.72* | *0.00* | *-2.09* | *+2.19* | *+0.06* |
| Olmo-3-7B-Think | 27.30 | 44.54 | 46.21 | 65.6 | 49.75 | 90.49 | 74.85 | 64.81 | 94.92 | 94.44 | 44.6 | 55.52 | 38.54 | - |
| + RL on HOW2TRAIN | 37.89 | 49.61 | 47.10 | 63.3 | 51.19 | 89.45 | 72.40 | 64.29 | 95.30 | 94.62 | 47.0 | 58.65 | 43.96 | - |
| *Delta* | *+10.59* | *+5.07* | *+0.89* | *-2.3* | *+1.44* | *-1.04* | *-2.45* | *-0.52* | *+0.38* | *+0.18* | *+2.4* | *+3.13* | *+5.42* | *+1.05* |

targeted analyses to stress-test each explanation, and find evidence that neither can account for the observed gains.

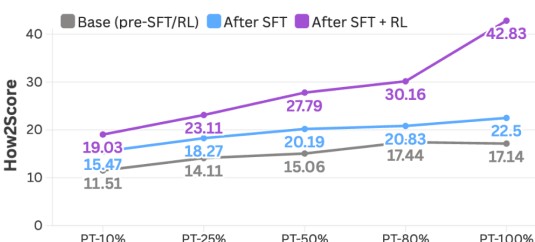

*Figure 4.* Post-training from different Olmo 3 7B pretraining checkpoints (x-axis). RL gains grow substantially at later checkpoints, while SFT yields modest improvements.

## 7.1. Confound 1: Implicit Task Format

To test whether gains from post-training can be explained by learning an implicit task format, we use two complementary diagnostics. **Pretraining maturity axis:** we hold the post-training recipe fixed and vary the pretraining checkpoint. If gains were primarily format-level, they should be similarly recoverable from weaker checkpoints. **Data coverage axis:** we hold the base model fixed and run RL using topic-restricted training data. If gains primarily come from format, they should transfer broadly across topics, largely independent of which topics appear in training.

**Diagnostic 1: Gains require pretraining maturity.** We apply the same post-training recipe (SFT followed by RL) starting from varying intermediate Olmo 3 7B pretraining checkpoints. Figure 4 shows that SFT gains are similar across checkpoints (3.39 to 5.36),[11] while RL gains increase with pretraining FLOPs (3.56 at 10% to 20.33 at 100%), accounting for most of the improvement at late checkpoints. This pattern aligns with prior findings that SFT mainly shapes surface-level behavior, while RL amplifies pretrained capabilities (Ouyang et al., 2022; Zhao et al., 2025), suggesting that HOW2BENCH is not primarily format-driven.

---

[11]Unlike §6.2, the SFT step yields additional performance improvement over RL alone when starting from base models.

**Diagnostic 2: Gains depend on data topic coverage.** Next, we test whether improvements depend on broad topic coverage or can be obtained by learning a generic output format from a narrow topic. Via embedding analysis, we select two topics with contrasting diversity: *Science, Math & Technology*, which is broadly dispersed in embedding space, and *Food & Dining*, which forms a specialized cluster. We run RL on Qwen 3 8B and find that training on all topics yields the best overall performance (+10.10), while science-only RL generalizes strongly to many other topics (+9.41 overall); in contrast, dining-only RL still transfers but more weakly (+5.55 overall). Full results in §E.6. Together, these results suggest that RL trained on a single topic can transfer, but broad topic coverage yields the largest gains, consistent with improvements driven by content coverage rather than a generic output format.

*Table 3.* **Midtraining memorization sensitivity.** As document occurrence increases, perplexity drops sharply, while HOW2SCORE only improves modestly and non-monotonically.

| Model | Metric ↓ | Doc occurrences during midtraining | | | | |
|---|---|---|---|---|---|---|
| | | 0 | 1 | 3 | 6 | 10 |
| Olmo 3 7B | Doc perplexity | 10.4 | 8.5 | 6.1 | 3.0 | 1.4 |
| | HOW2SCORE | 14.0 | 17.3 | 15.8 | 15.7 | 16.5 |
| Olmo 3 32B | Doc perplexity | 8.0 | 6.0 | 3.5 | 1.4 | 1.2 |
| | HOW2SCORE | 33.3 | 39.3 | 39.4 | 38.1 | 37.9 |

## 7.2. Confound 2: Memorization of Source Documents

To probe memorization effects, we vary how often a fixed set of source documents appears during midtraining and measure HOW2SCORE on procedure instances extracted from those documents.

**Midtraining.** We focus on *midtraining*, a stage where benchmark leakage can spuriously boost task scores (Olmo et al., 2025). Starting from the final pretraining checkpoints of Olmo 3 7B and 32B, we midtrain for 10B tokens while controlling for the exposure frequency of the documents (0, 1, 3, 6, or 10 occurrences). The 0-occurrence control serves as a baseline where the target documents are not seen.

**Memorizing source documents yields limited gains on HOW2SCORE.** We run our pipeline on the midtraining documents to create an evaluation set of 13,500 examples, evenly balanced across the five occurrence groups. Table 3 shows HOW2SCORE of our midtrained models on this evaluation set. As exposure increases, document perplexity drops sharply (for Olmo 3 7B: 10.4 → 1.4; for Olmo 3 32B: 8.0 → 1.2), indicating substantially higher fit to the source documents. By contrast, HOW2SCORE improves only modestly and non-monotonically (peaking at +3.3 for 7B and +6.1 for 32B), suggesting that improvements in HOW2SCORE are not explained simply by repeatedly seeing the underlying source documents.

## Impact Statement

We introduce HOW2EVERYTHING, a framework for mining and evaluating goal-conditioned step-by-step procedures from large-scale web corpora. Since step-by-step procedure generation is an important and commonly used capability of LLMs, providing measurement and data for improving it has practical and broad positive impacts. Our work enables more reliable evaluation of procedural instruction quality at scale, and provides a practical reward signal for improving models' end-to-end procedural validity. Altogether, it could benefit user-facing assistants in domains such as troubleshooting, education, and everyday planning.

**Risks and negative societal impact.** Because our data are derived from web documents, the extracted procedures may reflect societal biases present online. In addition, "how-to" instructions can be safety-sensitive (e.g., health, legal, chemicals), and misuse could enable harmful behavior if models are trained to generate unsafe instructions.

**Mitigations.** HOW2SCORE is an evaluation proxy rather than a guarantee of real-world correctness; it is not a substitute for expert review or execution-based verification in safety-critical settings. Prior to release, we will apply additional safety and privacy filtering (e.g., removing procedures that involve regulated or high-risk activities and removing personally identifiable information where present) and provide documentation describing intended use and known limitations. We will release the benchmark split, prompts, and distilled judge weights to support reproducible evaluation without requiring access to proprietary judge models.

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

# A. Motivational analysis of query type distribution

Chatterji et al. (2025) reports that *How-To Advice* accounts for approximately 8.5% of conversations for ChatGPT usage, ranking 4th among 23 fine-grained categories. *How-To Advice* comes right behind *Specific Info* (18.3%), *Edit or Critique Provided Text* (10.6%), and *Tutoring or Teaching* (10.2%). At current ChatGPT scale, this corresponds to tens to hundreds of millions of how-to interactions daily.

We also apply the same query-type classifier to two publicly available corpora, WildChat-4.8M (Zhao et al., 2024) and LMSYS-Chat-1M (Zheng et al., 2024), but observe systematic skews in query-type distributions relative to ChatGPT, consistent with these corpora being collected from unrestricted public LLM endpoints rather than from a deployed product. Accordingly, we anchor our discussion of real-world LLM usage to the ChatGPT distribution and use the open datasets primarily to contextualize results under alternative user behavior regimes.

See Table 4 for a summary of the query type distribution across ChatGPT, LMSYS, and WildChat.

*Table 4.* Query type distribution across chat sources (percent of conversations). OpenAI numbers are taken from Chatterji et al. (2025); LMSYS and WildChat are computed by applying the same classifier rubric used in the OpenAI report to LMSYS-Chat-1M (Zheng et al., 2024) and WildChat-4.8M (Zhao et al., 2024). We visually emphasize the OpenAI (commercial) column.

| Query type | OpenAI | LMSYS | WildChat |
|---|---|---|---|
| Specific Info | **18.3%** | 13.5% | 8.7% |
| Edit or Critique Provided Text | **10.6%** | 6.5% | 13.2% |
| Tutoring or Teaching | **10.2%** | 7.1% | 7.0% |
| How To Advice | **8.5%** | 5.5% | 3.4% |
| Personal Writing or Communication | **8.0%** | 7.4% | 4.0% |
| Health, Fitness, Beauty or Self Care | **5.7%** | 1.4% | 1.1% |
| Translation | **4.5%** | 1.4% | 6.4% |
| Computer Programming | **4.2%** | 13.2% | 8.8% |
| Create an Image | **4.2%** | 0.6% | 6.1% |
| Other / Unknown | **4.1%** | 3.1% | 5.2% |
| Creative Ideation | **3.9%** | 3.1% | 4.7% |
| Argument or Summary Generation | **3.6%** | 7.5% | 5.5% |
| Mathematical Calculation | **3.0%** | 3.0% | 1.5% |
| Purchasable Products | **2.1%** | 0.8% | 0.7% |
| Greetings and Chitchat | **2.0%** | 5.8% | 8.1% |
| Relationships and Personal Reflection | **1.9%** | 1.5% | 0.8% |
| Write Fiction | **1.4%** | 7.1% | 6.0% |
| Generate or Retrieve Other Media | **1.1%** | 0.2% | 0.2% |
| Cooking and Recipes | **0.9%** | 0.6% | 0.5% |
| Analyze an Image | **0.6%** | 0.2% | 0.1% |
| Data Analysis | **0.4%** | 1.9% | 5.2% |
| Games and Role Play | **0.4%** | 3.8% | 1.7% |
| Asking About the Model | **0.4%** | 4.7% | 1.1% |

# B. Details on data pipeline

This section provides details related to the data. Prompts used for the data pipeline are provided in subsection H.1.

## B.1. Top frequent URL domains

See Table 5 for the top 10 most frequent URL domains *within each topic* in our final pool of 351K procedure instances. Counts are the number of procedure instances in the given topic whose source URL belongs to the domain.

*Table 5.* Top 10 most frequent URL domains per topic in our final pool of 351K procedure instances.

| Rank | Domain | Count |
|---|---|---|
| **Art & Design** | | |
| 1 | instructables.com | 237 |
| 2 | creativelive.com | 212 |
| 3 | steves-digicams.com | 126 |
| 4 | shutterbug.com | 97 |
| 5 | dummies.com | 95 |
| 6 | wikihow.com | 81 |
| 7 | picturecorrect.com | 73 |
| 8 | digital-photography-school.com | 68 |
| 9 | snapshot.canon-asia.com | 67 |
| 10 | photography.tutsplus.com | 67 |
| **Crime & Law** | | |
| 1 | wikihow.com | 414 |
| 2 | legalbeagle.com | 241 |
| 3 | policeone.com | 159 |
| 4 | policemag.com | 88 |
| 5 | patternlanguagenetwork.org | 79 |
| 6 | info.legalzoom.com | 77 |
| 7 | avvo.com | 76 |
| 8 | americanbar.org | 70 |
| 9 | nolo.com | 60 |
| 10 | insidecounsel.com | 54 |
| **Education & Jobs** | | |
| 1 | betterlesson.com | 201 |
| 2 | wikihow.com | 128 |
| 3 | englishlessonplanner.com | 124 |
| 4 | classroom.synonym.com | 95 |
| 5 | work.chron.com | 95 |
| 6 | auburn.edu | 85 |
| 7 | education.com | 82 |
| 8 | slideplayer.com | 64 |
| 9 | prezi.com | 52 |
| 10 | brighthubeducation.com | 50 |
| **Electronics & Hardware** | | |
| 1 | instructables.com | 1348 |
| 2 | wikihow.com | 121 |
| 3 | lifehacker.com | 114 |
| 4 | ecmweb.com | 76 |
| 5 | dummies.com | 73 |
| 6 | hackaday.com | 68 |
| 7 | crutchfield.com | 53 |
| 8 | hackaday.io | 49 |
| 9 | itstillworks.com | 49 |
| 10 | lifewire.com | 49 |
| **Fashion & Beauty** | | |
| 1 | wikihow.com | 753 |
| 2 | leaf.tv | 437 |
| 3 | reference.com | 347 |
| 4 | oureverydaylife.com | 260 |
| 5 | instructables.com | 229 |
| 6 | naturallycurly.com | 186 |
| 7 | allure.com | 181 |
| 8 | popsugar.com | 171 |
| 9 | becomegorgeous.com | 137 |
| 10 | cosmopolitan.com | 136 |
| **Food & Dining** | | |
| 1 | recipe-finder.com | 1065 |
| 2 | food.com | 817 |
| 3 | ifood.tv | 569 |
| 4 | dlife.com | 385 |
| 5 | foodandwine.com | 313 |
| 6 | instructables.com | 244 |
| 7 | vegweb.com | 242 |
| 8 | seriouseats.com | 241 |
| 9 | relish.com | 234 |
| 10 | washoku.guide | 210 |

*Continued on next page.*

| Rank | Domain | Count |
|---|---|---|
| **Health** | | |
| 1 | wikihow.com | 370 |
| 2 | healthyliving.azcentral.com | 87 |
| 3 | infobarrel.com | 85 |
| 4 | slideplayer.com | 82 |
| 5 | livestrong.com | 63 |
| 6 | dummies.com | 47 |
| 7 | lifehacker.com | 44 |
| 8 | futurelearn.com | 44 |
| 9 | leaf.tv | 44 |
| 10 | hubpages.com | 42 |
| **Home & Hobbies** | | |
| 1 | instructables.com | 1083 |
| 2 | homeguides.sfgate.com | 787 |
| 3 | wikihow.com | 359 |
| 4 | hunker.com | 291 |
| 5 | reference.com | 216 |
| 6 | homesteady.com | 199 |
| 7 | ehow.com | 183 |
| 8 | doityourself.com | 173 |
| 9 | thespruce.com | 169 |
| 10 | lifehacker.com | 160 |
| **Industrial** | | |
| 1 | ecmweb.com | 280 |
| 2 | forconstructionpros.com | 195 |
| 3 | ptonline.com | 138 |
| 4 | thefabricator.com | 116 |
| 5 | machinerylubrication.com | 116 |
| 6 | screenweb.com | 110 |
| 7 | beefmagazine.com | 106 |
| 8 | instructables.com | 104 |
| 9 | weldingtipsandtricks.com | 103 |
| 10 | wikihow.com | 100 |
| **Religion** | | |
| 1 | lds.org | 610 |
| 2 | uua.org | 146 |
| 3 | wikihow.com | 131 |
| 4 | spellsofmagic.com | 117 |
| 5 | classroom.synonym.com | 105 |
| 6 | bible.org | 102 |
| 7 | orthodoxsundayschool.org | 79 |
| 8 | childrensministry.com | 78 |
| 9 | ssnet.org | 77 |
| 10 | teachonereachone.org | 59 |
| **Science, Math & Technology** | | |
| 1 | education.com | 361 |
| 2 | slideplayer.com | 297 |
| 3 | instructables.com | 284 |
| 4 | getrevising.co.uk | 208 |
| 5 | nrich.maths.org | 192 |
| 6 | openwetware.org | 167 |
| 7 | betterlesson.com | 159 |
| 8 | sciencebuddies.org | 121 |
| 9 | ck12.org | 118 |
| 10 | prezi.com | 92 |
| **Sports & Fitness** | | |
| 1 | healthyliving.azcentral.com | 360 |
| 2 | wikihow.com | 329 |
| 3 | t-nation.com | 287 |
| 4 | bodybuilding.com | 249 |
| 5 | active.com | 231 |
| 6 | woman.thenest.com | 207 |
| 7 | livehealthy.chron.com | 178 |
| 8 | runnersworld.com | 176 |
| 9 | howcast.com | 163 |
| 10 | mensfitness.com | 162 |
| **Transportation** | | |
| 1 | hotrod.com | 260 |

| Source format | # docs | After extraction (%) | After heuristics (%) | After LLM filter (%) |
|---|---|---|---|---|
| Tutorial & how-to guide | 140,000 | 86.87% | 70.46% | 24.46% |
| Personal blog | 140,000 | 34.48% | 25.41% | 14.26% |
| Knowledge articles | 140,000 | 35.47% | 23.72% | 16.54% |
| Non-fiction writing | 140,000 | 33.65% | 24.80% | 13.65% |
| Q&A forum | 140,000 | 49.02% | 31.75% | 17.67% |
| Academic writing | 122,439 | 29.20% | 23.21% | 16.29% |

*Table 6.* Aggregate yield rates by WebOrganizer document format, measured through the procedure-extraction stage, heuristics filtering, and the LLM filter stage (before postprocessing and final validation). All yields are reported as a percentage of the original input source documents for each format.

| Rank | Domain | Count |
|---|---|---|
| 2 | itstillruns.com | 219 |
| 3 | wikihow.com | 204 |
| 4 | instructables.com | 184 |
| 5 | superchevy.com | 140 |
| 6 | ixigo.com | 122 |
| 7 | reference.com | 104 |
| 8 | popularmechanics.com | 100 |
| 9 | auto.howstuffworks.com | 96 |
| 10 | aopa.org | 87 |
| **Travel & Tourism** | | |
| 1 | traveltips.usatoday.com | 256 |
| 2 | ixigo.com | 168 |
| 3 | wikihow.com | 146 |
| 4 | tripsavvy.com | 75 |
| 5 | lifehacker.com | 61 |
| 6 | getawaytips.azcentral.com | 58 |
| 7 | frommers.com | 55 |
| 8 | budgettravel.com | 45 |
| 9 | cruisemates.com | 30 |
| 10 | instructables.com | 28 |

## B.2. Extracting procedures from documents of other formats

See Table 6 for yield rates for WebOrganizer document formats beyond "Tutorial & How-to Guide", measured through the procedure-extraction stage, heuristics filtering, and the LLM filter stage (before postprocessing and final validation). Throughout, *yield* at a stage is computed relative to the original input source documents for that format (i.e., #documents retained after the stage divided by #input documents). Overall, these results show that valid, extractable procedures are not unique to tutorial-style pages, but "Tutorial & How-to Guide" consistently achieves the highest yields at each stage, so we focus on it as the primary source format for efficient large-scale mining.

## B.3. Implementation details on the heuristics filter

We apply two simple filters in sequence: **(1) Step-count filter.** We require the extracted procedure to have between min_steps and max_steps steps (defaults: 5–15; configurable via command-line flags). **(2) N-gram repetition filter.** We normalize each step (lowercasing and removing punctuation) and compute the repetition rate of 2-, 3-, and 4-grams pooled across all steps, where repetition rate is the fraction of n-grams that are repeated beyond their first occurrence. We reject procedures with high repetition, using thresholds of $\geq 0.40$ for bigrams, $\geq 0.35$ for trigrams, or $\geq 0.30$ for fourgrams. This filter primarily removes degenerate extractions that repeat near-identical phrases or steps.

## B.4. Rationale behind LLM filter criteria

At the LLM filter stage in HOW2MINE, we remove procedures that fall within the following categories (matching the prompt in §H.1):

- **Named-entity focused.** These instances hinge on entity-specific conventions (a particular person, organization, website, software product, or brand). Their correctness is often time- and access-dependent, and cannot be judged reliably without the entity context.

- **Pure math.** Pure calculation or formula-solving tasks are not procedures in our intended sense: success is determined by a correct numeric/algebraic result and is better evaluated by mathematical or verifiable oracles rather than critical-failure detection in instructions.

- **UI interaction.** UI-driven tasks require interacting with specific interfaces (websites, apps, terminals) and implicit state (what is currently visible, which buttons exist, what menus are named). In our execution-free setting, these tasks are difficult to verify and are brittle to UI changes over time.

- **Open-ended creative generation.** Creative goals have many qualitatively different valid endpoints, and the boundary between valid and invalid is dominated by taste and preference rather than by missing prerequisites or contradictions.

- **Non-sequential process.** Some examples are practically listicles, where most steps are order-independent without clear linear dependencies. Such instances are ill-posed for step-by-step validity evaluation.

- **Unreasonable procedure.** Finally, we remove instances where the steps are internally inconsistent or cannot plausibly achieve the stated goal (e.g., contradictions, logically impossible actions, or missing essential actions). Some are also simply nonsensical (e.g., "How to fail an exam"). This serves as a quality-control stage: since HOW2SCORE uses the mined reference procedure as an anchor, unreasonable references would directly add noise to both evaluation and training.

## B.5. Example procedures for all 14 topics

In Table 7, we show one full example (goal, resources, reference steps) for each of the 14 topics.

*Table 7.* One full example (goal, resources, reference steps) for each of the 14 topics used in HOW2EVERYTHING.

| Topic | Goal | Resources | Reference steps |
|---|---|---|---|
| **Art & Design** | To produce a glass piece featuring a reticello network pattern using the process of forming, twisting, and combining two color-cored glass cups. | color-cored glass canes; glass collar | **1.** Form a cup by rolling color-cored glass canes around a glass collar. **2.** Twist the cup to create a spiral pattern. **3.** Remove the twisted cup from the collar and set it aside. **4.** Form a second cup by rolling color-cored glass canes around a glass collar. **5.** Twist the second cup in the opposite direction to the first cup. **6.** Blow the second cup inside the first cup to create the reticello network pattern. |
| **Crime & Law** | Sell your share of a common property apartment with separate ownership by following the required legal procedure for notifying co-owners and transferring ownership. | notary; notarial document; letter with a list of contents; receipt | **1.** Prepare a notification to all co-owners stating the conditions of sale of your share. **2.** Visit a notary to draw up a notarial document including all sale conditions. **3.** Distribute the notarial document to all co-owners by letter with a list of contents, obtaining a receipt from each. **4.** Wait 30 days for co-owners to express their desire to purchase your share. **5.** Sell your share to a third party. |
| **Education & Jobs** | To create an it-cleft sentence in English that emphasizes a specific part of a given simple sentence. | simple sentence; BE verb ("was" or "is"); "It"; "who"; "that"; "when" | **1.** Choose the simple sentence to transform. **2.** Select the part of the sentence to emphasize (subject, object, or time). **3.** Start the new sentence with "It" and the appropriate form of the BE verb ("was" or "is"). **4.** Place the chosen part to emphasize immediately after the BE verb. **5.** Insert "who," "that," or "when" as appropriate to introduce the rest of the sentence. **6.** Add the remaining information from the original sentence to complete the it-cleft sentence. |

*Continued on next page.*

| Topic | Goal | Resources | Reference steps |
|---|---|---|---|
| Electronics & Hardware | To maintain a pedestrian turnstile gate to ensure its proper functioning and extend its service life through a regular, comprehensive maintenance procedure. | soft cloth; vacuum cleaner; lubricant; antirust oil; stainless steel maintenance oil; paint; non-corrosive cleaning solution; soft lint-free rag | **1.** Cut off the power supply to the turnstile gate. **2.** Open the cover of the turnstile gate chassis. **3.** Clean dust and debris from the surface and interior using a soft cloth or vacuum cleaner. **4.** Tighten any loose connecting screws on all internal parts. **5.** Apply lubricant to moving components after inspecting the wear of vulnerable parts. **6.** Adjust the balance spring after 30,000 operations. **7.** Replace any aging or damaged wires in the power circuit. **8.** Polish the external chassis with a soft cloth and apply antirust oil or stainless steel maintenance oil. **9.** Repair exposed scratches on the chassis with paint of the same color. **10.** Clean the infrared acrylic mirror and beam window with a non-corrosive cleaning solution and a soft lint-free rag. |
| Food & Dining | Prepare fricasé Boliviano, a spicy pork stew with potatoes and white corn, by cooking pork with spices, thickening the stew, and serving with cooked potatoes and white corn. | oil; large pot; pork pieces; white onion; cumin; black pepper; garlic; cayenne pepper; green onion; water; pan; potatoes; white corn; bread crumbs; deep plate | **1.** Heat oil in a large pot. **2.** Fry pork pieces in the oil until golden. **3.** Add white onion, cumin, black pepper, garlic, cayenne pepper, and green onion to the pot. **4.** Pour water into the pot while stirring. **5.** Simmer until the meat comes off the bones, maintaining the broth level as needed (about 2 hours). **6.** Cook potatoes in a separate pan until done. **7.** Cook white corn in a separate pan until done. **8.** Add bread crumbs to the stew to thicken it shortly before serving. **9.** Serve the stew in a deep plate and garnish with the cooked potatoes and white corn. |
| Fashion & Beauty | Create the structured base of an Uzbeki Spy Hat (or Wizard Hat) using interfacing and fabric. | interfacing; fabric; scissors; needle; thread | **1.** Cut a right triangle from the interfacing by folding one corner to meet the opposite side and cutting along the fold. **2.** Fold the triangle in half and stitch from the point downwards to form a cone. **3.** Trim the cone so it fits properly above your eyes. **4.** Lay the cone on your fabric with the seam next to one edge and cut around the bottom, leaving about an inch of seam allowance. **5.** Roll the interfacing cone over towards the adjacent side of the fabric to cut the other half. **6.** Stitch a cone of fabric. **7.** Trim the tips and turn both cones inside out. **8.** Fit the interfacing cone inside the fabric cone. **9.** Stitch the cones together near the base of the interfacing. |
| Health | To prevent the formation of venous ulcers in your legs by following a daily care routine. | compression stockings; lotion; antiseptic ointment | **1.** Wear compression stockings every day while you are awake. **2.** Exercise regularly to lose weight and lower blood pressure. **3.** Apply lotion to your legs every day. **4.** Check your legs for hard or rough areas and small cuts or abrasions while applying lotion. **5.** Use antiseptic ointment on every small sore. |
| Home & Hobbies | To create a knurled band on the face of a prop driver using the plunge knurling technique on a lathe. | lathe; knurl holder; tool post holder; peg; cross slide | **1.** Reduce the stock to the final diameter required for the driver. **2.** Counter-bore to a depth of about 0.016 inches to produce the band that will be knurled. **3.** Mount the knurl holder in the tool post holder with the center of the peg set to lathe center height. **4.** Start the lathe and run the work at about 500 rpm. **5.** Plunge the knurl into the face of the drive washer to form the knurl. **6.** Run the cross slide in and out by about 1/32 inch to help clear chips and form the V's. **7.** Take a light skimming cut over the outside diameter of the driver to remove the metal burr. **8.** Take a light skimming cut on the inside of the band. |
| Industrial | Install a comprehensive waterproofing system for below grade spaces to prevent water ingress and structural damage. | high performance waterproofing; waterproofing membrane; comprehensive waterproofing system; installers; manufacturer; geotechnical report | **1.** Specify high performance waterproofing suitable for the assessed risk. **2.** Ensure the waterproofing membrane bonds adhesively to the structure to prevent lateral water migration. **3.** Specify a comprehensive waterproofing system for both floors and walls. **4.** Confirm that installers are experienced and trained by the manufacturer, and arrange for manufacturer support such as preinstall meetings and site visits. **5.** Have the manufacturer review the geotechnical report to ensure membrane compatibility with site contaminants. |

| Topic | Goal | Resources | Reference steps |
|---|---|---|---|
| **Religion** | Analyze the influence of decanates and their associated Areas of Consciousness on the principal theme of a natal chart. | natal chart; decanates; Areas of Consciousness | **1.** Determine the decanates occupied by the Sun, Moon, Ascendant, Ruling Planet, and ruler of the 5th house in the natal chart. **2.** Identify which decanate (first, second, or third) is most frequently occupied by the majority of these key points. **3.** Associate the most emphasized decanate with its corresponding Area of Consciousness: Personal (first), Relating (second), or Universal (third). **4.** Highlight any key points not in the majority decanate for special consideration regarding their expression. **5.** Integrate the decanate emphasis and associated Areas of Consciousness with the principal theme of the natal chart. |
| **Science, Math & Technology** | Compute the trajectory of a particle through a velocity field using numerical integration within a grid. | cell; velocity field; grid; Euler's method | **1.** Identify the cell containing the initial position of the particle. **2.** Determine the velocity at the current position by interpolation. **3.** Calculate the new position using Euler's method. **4.** Identify the cell containing the new position. **5.** Repeat the previous three steps while the particle remains inside the grid. |
| **Sports & Fitness** | Complete a specific yoga sequence designed to stretch and strengthen the core muscles for equestrian fitness. | yoga mat | **1.** Practice three-part breath (pranayama), expanding the stomach, then ribs, then chest on each inhale. **2.** Move the spine in all directions—front, back, sides, and twists—while linking breath to movement. **3.** Hold Goddess pose by standing with legs wide, bending knees, keeping chest elevated and shoulders over hips, tucking tailbone, and sinking into the squat. **4.** Hold Warrior 1 pose by facing forward, stretching one leg back into a lunge, reaching both arms upward, keeping shoulders wide, and elongating the torso by drawing the belly button toward the spine; alternate legs and hold each side for 10 breaths. **5.** Hold Downward Facing Dog by bending down from standing, stretching legs back to high plank, then pressing hands down and lifting hips up and back to form an inverted "V", keeping weight evenly distributed between hands and feet. **6.** Lie on your back in Savasana (Corpse pose) and relax for several minutes. |
| **Transportation** | Check a car seat as baggage at the airport to minimize the risk of damage or loss by following a specific procedure. | large corrugated cardboard box; contact information; airline check-in counter; airline staff; luggage tag; claim ticket; baggage area; airline's baggage desk; photos | **1.** Pack the car seat securely in a large corrugated cardboard box. **2.** Label the box with your contact information. **3.** Bring the packed car seat to the airline check-in counter. **4.** Check the car seat as baggage with the airline staff and obtain a luggage tag or claim ticket. **5.** Inspect the car seat for visible damage upon arrival at your destination before leaving the baggage area. **6.** File a claim at the airline's baggage desk immediately if the car seat is lost or visibly damaged, providing the luggage tag information and photos. |
| **Travel & Tourism** | Verify the authenticity of a U.S. e-passport and the identity of its holder at a border control terminal using the e-passport chip and printed information. | passport terminal; e-passport chip; printed key; passport book; person presenting the passport | **1.** Unlock the e-passport chip using the printed key from the passport book. **2.** Establish communication between the passport terminal and the unlocked chip over a short distance. **3.** Transmit the encrypted data from the chip to the passport terminal. **4.** Verify the digital signature on the chip's data to confirm authenticity and detect tampering. **5.** Compare the printed information, the digital information from the chip, and the person presenting the passport. |

## C. Details on developing How2Score

### C.1. Human annotation setup

In the final version of annotator training, we first define critical failures, then carefully walk annotators through five examples of critical failures and six acceptable variances that should not be counted as critical.

Our annotation interface enforces attention checks: annotators must explicitly click through UI elements to confirm that they have read and understood each example. Submissions remain closed until at least 90 seconds have elapsed. See screenshots of our annotation interface in Figures 5, 6, 7, 8, 9.

### C.2. Distillation

To construct training data for distilling GPT 5 into a Qwen 3 8B judge, we collect 72,920 GPT 5 annotations on model-generated procedures from a diverse set of generator models. Specifically, we include generations from:

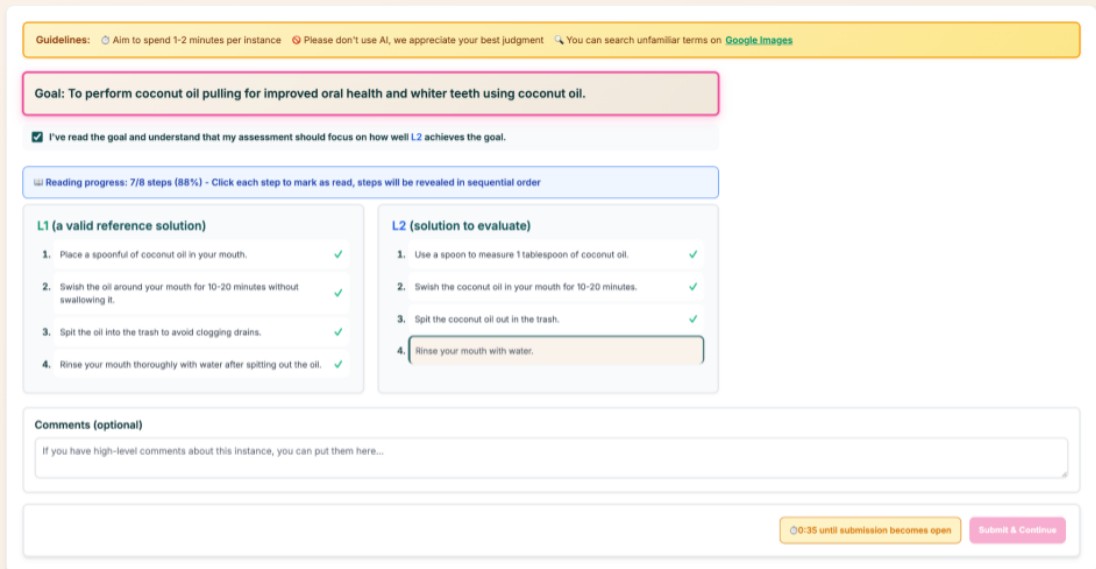

*Figure 5.* **Annotation interface screenshot [1].** Annotators must acknowledge they have read and understood the goal, then click through all model-generated steps to confirm they have thoroughly read them.

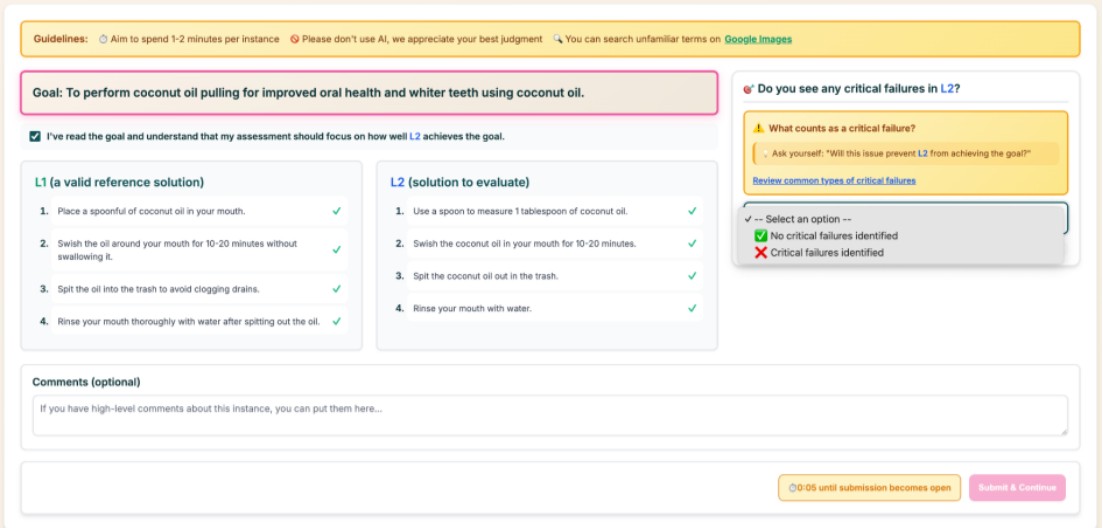

*Figure 6.* **Annotation interface screenshot [2].** After reading the example, annotators select whether they do identify a critical failure.

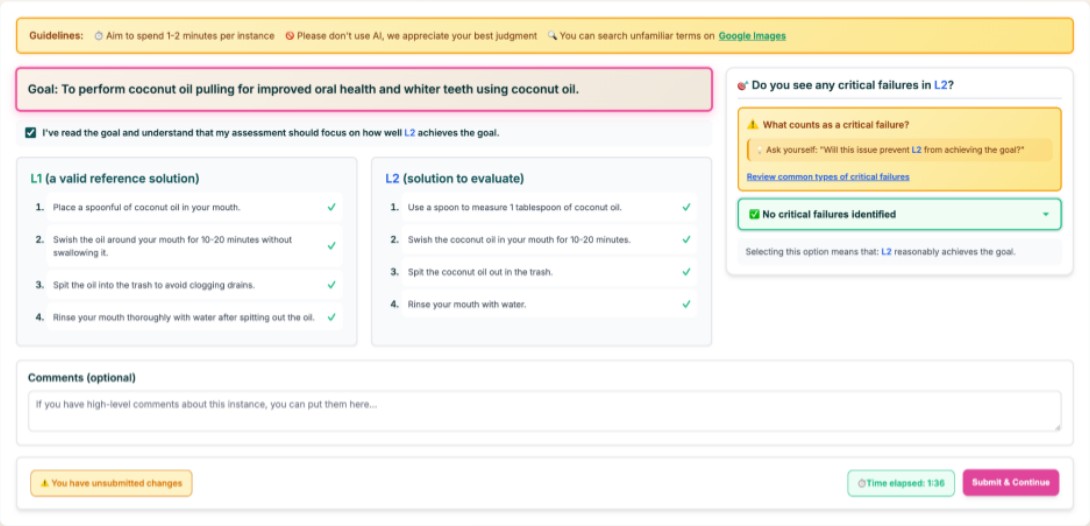

*Figure 7.* **Annotation interface screenshot [3].** If there is no critical failure, annotators can select that option from the dropdown, and submit.

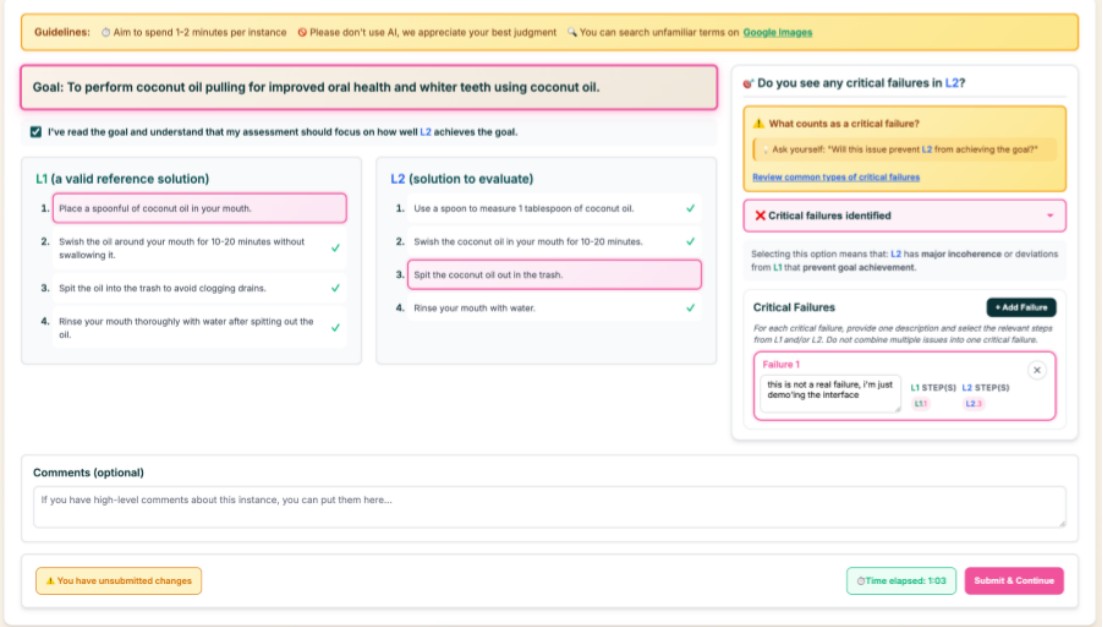

*Figure 8.* **Annotation interface screenshot [4].** If annotators do identify a critical failure, they need to provide a brief description, then click any relevant reference / generation steps.

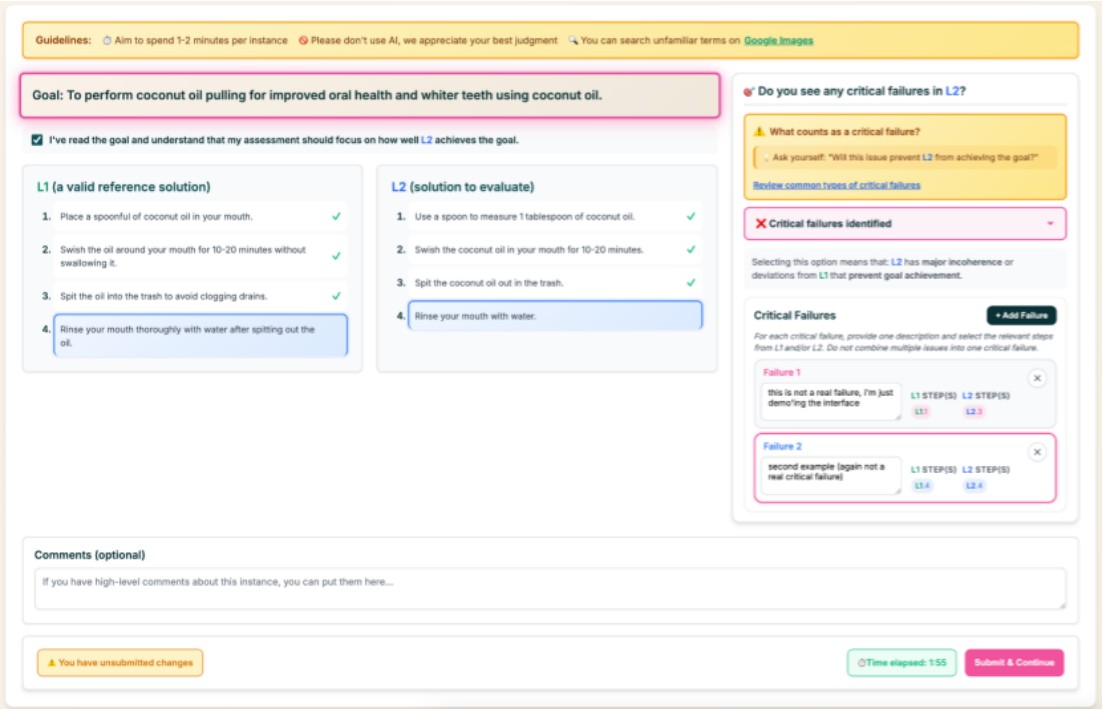

*Figure 9.* **Annotation interface screenshot [5].** Annotators can add multiple instances of critical failures.

- gemini-2.5-flash

- gemini-2.5-pro (Google 2025)

- gpt-4.1 (OpenAI 2025)

- GPT 5 (OpenAI 2026)

- qwen2.5-7b-instruct (Qwen Team 2024)

- OLMo-2-0425-1B-Instruct

- OLMo-2-0425-1B-stage1-step760000-tokens1594B

- OLMo-2-0425-1B-stage1-step1907359-tokens4001B

- OLMo-2-1124-7B-Instruct

- OLMo-2-1124-7B-stage1-step467000-tokens1959B

- OLMo-2-1124-7B-stage1-step928646-tokens3896B

- OLMo-2-0325-32B-Instruct

- OLMo-2-0325-32B-stage1-step467000-tokens3918B

- OLMo-2-0325-32B-stage1-step721901-tokens6056B

To reduce label noise from stochastic judging, we run GPT 5 twice for each example and retain only examples where the binary judgment (has_failure vs. no_failure) is consistent across both runs. We fine-tune the Qwen 3 8B judge for 3 epochs with learning rate $5e-6$ and batch size 64.

# D. Evaluation details

## D.1. Inference setup details

At inference time, the generator model receives the goal $g$, the resource list $R$, and the required step count $n = |S^\star|$, and is asked to output a procedure $\hat{S}$ with exactly $n$ steps.[12]

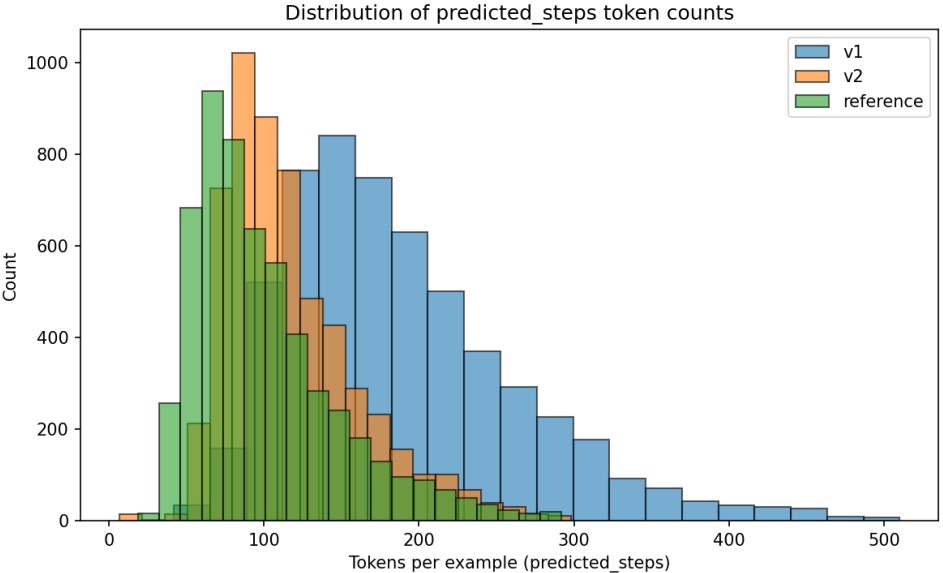

*Figure 10.* Requesting model to stick to the level detail in the few-shot examples in the inference prompt ("v2") significantly brings the generation length distribution closer to the reference distribution.

**Inference-time control for length bias.** Like many existing benchmarks that use LLM judges (Zheng et al., 2023; Dubois et al., 2025), HOW2BENCH shows length bias. In early experiments, we found that rewriting a procedure to be more verbose (while attempting not to introduce new information) can increase scores, even when the underlying procedural content is unchanged. To avoid unfairly rewarding verbosity, we standardize inference across all models using the same 3-shot prompts. Full prompts in §H.2. The prompts used for *base* and *instruct* endpoints are slightly different in wording. For instruction-tuned models, we append a more instruction-like suffix to the prompt. The examples in the prompt illustrate the expected output format and intended level of detail. We enforce explicit length control by requiring each step to be a single, concise sentence containing one main action, and asking the model to closely follow the concision level in the provided examples. Figure 10 shows that this explicit length control brings generated lengths substantially closer to the reference length distribution.

**Decoding setup.** For non-reasoning ("non-thinking") model endpoints, we use greedy decoding. To prevent overly long continuations (especially from base models), we use a stop sequence of \n\n for these endpoints. For reasoning-enabled API models, we use stochastic decoding with temperature $T = 0.6$, and use the provider's default reasoning/thinking budget for each API.

## D.2. Cross-judge robustness check for self-preference bias

See Figure 11 for results on the cross-judge robustness check. The generator models include GPT 5.1 and GPT 5 (OpenAI, 2026), Claude Opus 4.5 (Anthropic, 2025), Gemini 2.5 Pro (Google, 2025), and Gemini 3 Pro Preview (Google DeepMind, 2025). The judges include our distilled 8B judge, GPT 5 (OpenAI, 2026), Gemini 2.5 Pro (Google, 2025), and Claude 4.5 Opus (Anthropic, 2025).

---

[12]Conditioning generations on $R$ and $n$ is an evaluation control that reduces degrees of freedom and improves comparability across model outputs. It is not intended to reflect typical real-world usage.

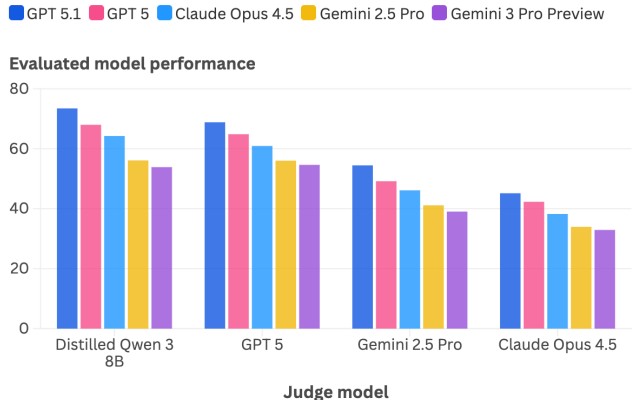

*Figure 11.* Cross-judge robustness check for self-preference bias on closed models spanning the GPT, Gemini, and Claude families: we rescore the same generations with four judges (distilled Qwen3 8B, GPT 5, Gemini 2.5 Pro, Claude 4.5 Opus) and find the ranking is unchanged.

### D.3. HOW2BENCH results on more models

See HOW2BENCH results on more models in Figure 12.

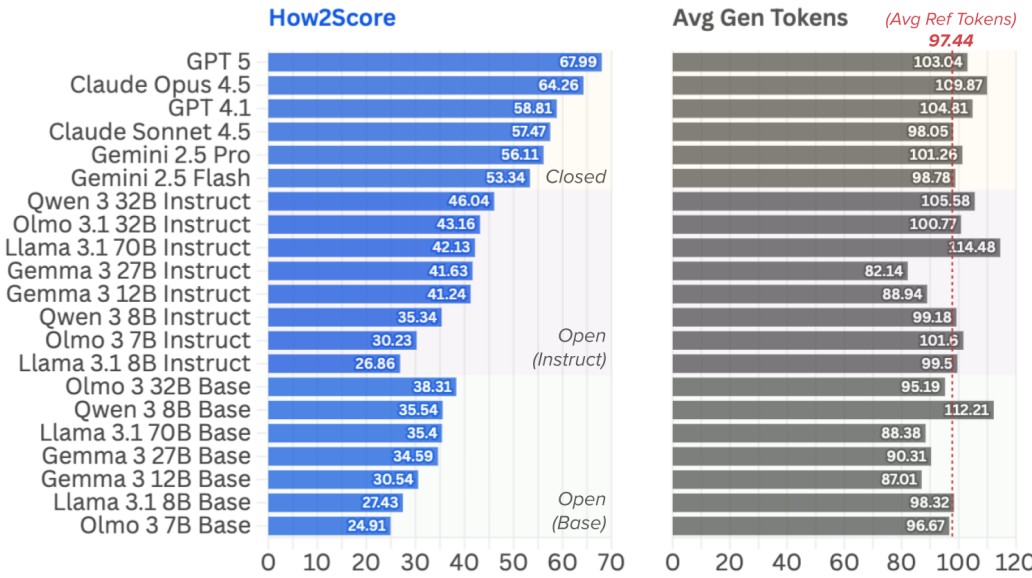

*Figure 12.* HOW2BENCH results on selected models. We report HOW2SCORE computed with our distilled judge along with the average generated tokens for each model. The average reference length is **97.44 tokens**. For open models, **Base** denotes the final non-post-trained checkpoint, and **Instruct** denotes the post-trained checkpoint.

### D.4. Full intermediate checkpoint evaluation results on HOW2BENCH

*Table 8.* Full checkpoint results on HOW2BENCH, including pretraining trajectories. For OLMo, *Pretrain* corresponds to the stage1 checkpoint series, *Midtrain* to the final base checkpoint, and *Posttrain* to the final instruct checkpoint.

| Suite | Size | Stage | Step | HOW2SCORE | Avg gen tokens |
|---|---|---|---|---|---|
| **OLMo-2** | | | | | |
| OLMo-2-0425 | 1B | Pretrain | 20000 | 0.06 | 247.74 |
| OLMo-2-0425 | 1B | Pretrain | 100000 | 0.56 | 109.12 |
| OLMo-2-0425 | 1B | Pretrain | 190000 | 0.76 | 105.18 |
| OLMo-2-0425 | 1B | Pretrain | 380000 | 0.80 | 86.24 |
| OLMo-2-0425 | 1B | Pretrain | 760000 | 0.96 | 101.86 |
| OLMo-2-0425 | 1B | Pretrain | 1140000 | 1.51 | 83.21 |
| OLMo-2-0425 | 1B | Pretrain | 1530000 | 1.49 | 82.71 |
| OLMo-2-0425 | 1B | Pretrain | 1907359 | 1.59 | 81.69 |
| OLMo-2-0425 | 1B | Midtrain | – | 6.39 | 82.63 |
| OLMo-2-0425 | 1B | Posttrain | – | 5.96 | 66.86 |
| OLMo-2-1124 | 7B | Pretrain | 9000 | 0.09 | 132.99 |
| OLMo-2-1124 | 7B | Pretrain | 46000 | 2.61 | 87.79 |
| OLMo-2-1124 | 7B | Pretrain | 93000 | 4.74 | 77.82 |
| OLMo-2-1124 | 7B | Pretrain | 187000 | 7.10 | 91.71 |
| OLMo-2-1124 | 7B | Pretrain | 371000 | 7.66 | 82.53 |
| OLMo-2-1124 | 7B | Pretrain | 557000 | 8.39 | 85.12 |
| OLMo-2-1124 | 7B | Pretrain | 743000 | 8.84 | 81.32 |
| OLMo-2-1124 | 7B | Pretrain | 928646 | 10.43 | 89.93 |
| OLMo-2-1124 | 7B | Midtrain | – | 22.29 | 90.20 |
| OLMo-2-1124 | 7B | Posttrain | – | 27.36 | 96.40 |
| OLMo-2-0325 | 32B | Pretrain | 7000 | 1.79 | 108.28 |
| OLMo-2-0325 | 32B | Pretrain | 36000 | 8.60 | 79.71 |
| OLMo-2-0325 | 32B | Pretrain | 72000 | 12.29 | 83.69 |
| OLMo-2-0325 | 32B | Pretrain | 145000 | 10.86 | 78.67 |
| OLMo-2-0325 | 32B | Pretrain | 289000 | 12.53 | 79.84 |
| OLMo-2-0325 | 32B | Pretrain | 433000 | 15.00 | 74.57 |
| OLMo-2-0325 | 32B | Pretrain | 578000 | 15.63 | 75.68 |
| OLMo-2-0325 | 32B | Pretrain | 721901 | 17.74 | 75.54 |
| OLMo-2-0325 | 32B | Midtrain | – | 35.50 | 94.94 |
| OLMo-2-0325 | 32B | Posttrain | – | 40.56 | 101.21 |
| **OLMo-3** | | | | | |
| OLMo-3-1025 | 7B | Pretrain | 14000 | 4.13 | 98.68 |
| OLMo-3-1025 | 7B | Pretrain | 71000 | 12.42 | 111.83 |
| OLMo-3-1025 | 7B | Pretrain | 141000 | 16.00 | 93.80 |
| OLMo-3-1025 | 7B | Pretrain | 283000 | 17.82 | 96.76 |
| OLMo-3-1025 | 7B | Pretrain | 566000 | 17.87 | 93.51 |
| OLMo-3-1025 | 7B | Pretrain | 848000 | 21.96 | 90.34 |
| OLMo-3-1025 | 7B | Pretrain | 1130000 | 21.46 | 90.85 |
| OLMo-3-1025 | 7B | Pretrain | 1413814 | 21.59 | 86.26 |
| OLMo-3-1025 | 7B | Midtrain | – | 24.91 | 96.67 |
| OLMo-3-1025 | 7B | Posttrain | – | 30.23 | 101.60 |
| OLMo-3-1125 | 32B | Pretrain | 6000 | 6.21 | 108.16 |
| OLMo-3-1125 | 32B | Pretrain | 29000 | 17.15 | 87.24 |
| OLMo-3-1125 | 32B | Pretrain | 58000 | 21.96 | 86.44 |
| OLMo-3-1125 | 32B | Pretrain | 116000 | 23.96 | 91.79 |
| OLMo-3-1125 | 32B | Pretrain | 232000 | 25.53 | 89.01 |
| OLMo-3-1125 | 32B | Pretrain | 347000 | 26.94 | 80.56 |
| OLMo-3-1125 | 32B | Pretrain | 463000 | 30.86 | 93.29 |
| OLMo-3-1125 | 32B | Pretrain | 579120 | 31.00 | 97.52 |
| OLMo-3-1125 | 32B | Midtrain | – | 38.31 | 95.19 |
| OLMo-3-1125 | 32B | Posttrain | – | 43.16 | 100.77 |

## D.5. Formatting proxy metrics over intermediate checkpoints

To complement HOW2EVERYTHING scores with simple automated checks of procedural *formatting*, we compute three proxy metrics on model generations across checkpoint trajectories and report them in Table 9. *Step-count mismatch* is the fraction of examples where the generated procedure has a different number of steps than the reference, i.e., $|\texttt{predicted\_steps}| \neq |\texttt{reference\_steps}|$. *Duplicate steps* is the fraction of examples where $\texttt{predicted\_steps}$ contains any exact repeated step string (verbatim duplicates), i.e., $|\texttt{set(predicted\_steps)}| \neq |\texttt{predicted\_steps}|$. *Dup n-gram rate* is computed within each example by concatenating $\texttt{predicted\_steps}$, whitespace-tokenizing, forming n-grams, and computing

$$\frac{\sum_g \max(0, c_g - 1)}{\text{total n-grams}},$$

where $c_g$ is the count of n-gram $g$; we then average over examples. In this table we report the unweighted mean across $n \in \{1, 2, 3, 4\}$.

*Table 9.* Formatting proxy metrics computed on model outputs across checkpoint trajectories (all values are percentages).

| Suite | Size | Stage | Step | Task score | Step-count mismatch | Dup-step ex. | Dup n-gram rate (1–4) |
|---|---|---|---|---|---|---|---|
| | | | | **OLMo-3-1125** | | | |
| OLMo-3-1125 | 32B | Pretrain | 7000 | 4.33% | 9.37% | 2.14% | 11.74% |
| OLMo-3-1125 | 32B | Pretrain | 33000 | 13.29% | 2.41% | 0.79% | 9.90% |
| OLMo-3-1125 | 32B | Pretrain | 66000 | 17.10% | 1.24% | 1.27% | 10.71% |
| OLMo-3-1125 | 32B | Pretrain | 131000 | 21.52% | 2.20% | 0.77% | 9.90% |
| OLMo-3-1125 | 32B | Pretrain | 262000 | 22.33% | 2.14% | 0.57% | 10.17% |
| OLMo-3-1125 | 32B | Pretrain | 394000 | 25.23% | 1.41% | 0.27% | 9.74% |
| OLMo-3-1125 | 32B | Pretrain | 525000 | 29.14% | 2.07% | 0.30% | 10.70% |
| OLMo-3-1125 | 32B | Pretrain | 656000 | 32.30% | 0.60% | 0.26% | 10.43% |
| OLMo-3-1125 | 32B | Midtrain | – | 35.23% | 1.30% | 0.16% | 9.42% |
| OLMo-3.1 | 32B | Posttrain | – | 42.47% | 1.71% | 0.00% | 8.99% |
| | | | | **OLMo-3-1025** | | | |
| OLMo-3-1025 | 7B | Pretrain | 14000 | 2.12% | 22.47% | 2.83% | 13.42% |
| OLMo-3-1025 | 7B | Pretrain | 71000 | 8.15% | 18.46% | 1.66% | 11.50% |
| OLMo-3-1025 | 7B | Pretrain | 141000 | 11.55% | 6.73% | 0.66% | 10.19% |
| OLMo-3-1025 | 7B | Pretrain | 283000 | 13.44% | 6.94% | 1.97% | 11.13% |
| OLMo-3-1025 | 7B | Pretrain | 566000 | 13.53% | 7.01% | 2.56% | 11.77% |
| OLMo-3-1025 | 7B | Pretrain | 848000 | 17.91% | 3.63% | 1.66% | 10.61% |
| OLMo-3-1025 | 7B | Pretrain | 1130000 | 17.44% | 3.81% | 1.50% | 10.79% |
| OLMo-3-1025 | 7B | Pretrain | 1413814 | 17.19% | 3.37% | 1.47% | 10.14% |
| OLMo-3-1025 | 7B | Midtrain | – | 21.51% | 1.97% | 0.24% | 9.72% |
| OLMo-3 | 7B | Posttrain | – | 29.80% | 0.03% | 0.00% | 9.06% |
| | | | | **OLMo-2-0425** | | | |
| OLMo-2-0425 | 1B | Pretrain | 20000 | 0.01% | 59.61% | 35.44% | 36.98% |
| OLMo-2-0425 | 1B | Pretrain | 100000 | 0.21% | 19.66% | 12.63% | 21.41% |
| OLMo-2-0425 | 1B | Pretrain | 190000 | 0.50% | 23.11% | 8.16% | 18.91% |
| OLMo-2-0425 | 1B | Pretrain | 380000 | 0.34% | 11.96% | 9.29% | 20.29% |
| OLMo-2-0425 | 1B | Pretrain | 760000 | 0.57% | 16.26% | 7.49% | 18.81% |
| OLMo-2-0425 | 1B | Pretrain | 1140000 | 0.56% | 15.81% | 5.11% | 16.06% |
| OLMo-2-0425 | 1B | Pretrain | 1530000 | 0.75% | 13.31% | 5.81% | 17.30% |
| OLMo-2-0425 | 1B | Pretrain | 1907359 | 1.02% | 12.14% | 5.59% | 17.18% |
| OLMo-2-0425 | 1B | Midtrain | – | 3.47% | 10.43% | 0.89% | 10.62% |
| OLMo-2-0425 | 1B | Posttrain | – | 4.34% | 36.99% | 0.01% | 8.47% |

| Suite | Size | Stage | Step | Task score | Step-count mismatch | Dup-step ex. | Dup n-gram rate (1–4) |
|---|---|---|---|---|---|---|---|
| | | | | | **OLMo-2-1124** | | |
| OLMo-2-1124 | 7B | Pretrain | 9000 | 0.04% | 37.67% | 18.57% | 26.40% |
| OLMo-2-1124 | 7B | Pretrain | 46000 | 1.76% | 7.17% | 5.64% | 16.67% |
| OLMo-2-1124 | 7B | Pretrain | 93000 | 2.88% | 4.77% | 4.30% | 16.42% |
| OLMo-2-1124 | 7B | Pretrain | 187000 | 4.30% | 8.53% | 4.83% | 16.01% |
| OLMo-2-1124 | 7B | Pretrain | 371000 | 5.07% | 6.90% | 4.10% | 16.36% |
| OLMo-2-1124 | 7B | Pretrain | 557000 | 5.97% | 7.81% | 4.91% | 16.49% |
| OLMo-2-1124 | 7B | Pretrain | 743000 | 6.47% | 10.10% | 4.27% | 15.69% |
| OLMo-2-1124 | 7B | Pretrain | 928646 | 6.57% | 10.23% | 3.81% | 15.08% |
| OLMo-2-1124 | 7B | Midtrain | – | 17.91% | 5.71% | 0.23% | 8.12% |
| OLMo-2-1124 | 7B | Posttrain | – | 27.62% | 0.10% | 0.00% | 9.21% |
| | | | | | **OLMo-2-0325** | | |
| OLMo-2-0325 | 32B | Pretrain | 7000 | 1.13% | 11.41% | 4.46% | 17.77% |
| OLMo-2-0325 | 32B | Pretrain | 36000 | 6.02% | 3.67% | 4.73% | 16.03% |
| OLMo-2-0325 | 32B | Pretrain | 72000 | 9.25% | 5.31% | 5.43% | 16.98% |
| OLMo-2-0325 | 32B | Pretrain | 145000 | 8.85% | 4.34% | 4.44% | 17.46% |
| OLMo-2-0325 | 32B | Pretrain | 289000 | 9.36% | 5.57% | 4.31% | 15.92% |
| OLMo-2-0325 | 32B | Pretrain | 433000 | 12.46% | 11.56% | 4.00% | 15.67% |
| OLMo-2-0325 | 32B | Pretrain | 578000 | 12.55% | 3.63% | 3.40% | 14.93% |
| OLMo-2-0325 | 32B | Pretrain | 721901 | 15.00% | 3.33% | 3.86% | 15.54% |
| OLMo-2-0325 | 32B | Midtrain | – | 32.25% | 2.73% | 0.17% | 8.26% |
| OLMo-2-0325 | 32B | Posttrain | – | 40.14% | 0.10% | 0.01% | 10.29% |

## D.6. Conditional perplexity vs. How2Score

We compute conditional perplexity (teacher-forced) on the *reference steps only*, conditioned on the goal and resources prompt for each example, and compare checkpoint ordering under this metric to checkpoint ordering under How2Score. Table 15 reports per-checkpoint How2Score scores and conditional reference-step perplexities, along with the induced within-run ranks. For each OLMo trajectory, the table header reports the Spearman rank correlation across the 9 checkpoints (8 stage-1 pretraining checkpoints plus the stage-2 midtrained checkpoint).

## D.7. Instance-level correlates of How2Score no_failure

This section analyzes how the How2Score label (no_failure vs. has_failure) varies with three simple, instance-level properties: **the reference step count** $|S^\star|$ (which also determines the requested number of generated steps in our inference setup), **the resource count** $|R|$ (the number of resources extracted from the reference procedure and provided as part of the task specification), and **a generation-to-reference length ratio** that captures residual verbosity relative to the reference. We focus on 7 models: two open 7–8B models, two open 32B models, and three closed frontier models. Because topics differ in typical reference step counts and resource-list sizes, we fit a logistic regression with topic fixed effects that predicts the per-example How2Score binary label from these covariates:

$$\text{logit}(p(\texttt{no\_failure})) = \log \frac{p(\texttt{no\_failure})}{1 - p(\texttt{no\_failure})}$$

$$= \beta_0 + \beta_{\text{steps}} \cdot |S^\star| + \beta_{\text{res}} \cdot |R| + \beta_{\text{ratio}} \cdot \rho + \sum_{t \in \mathcal{T} \setminus \{t_0\}} \gamma_t \, \mathbb{I}[\text{topic} = t], \quad (1)$$

where $|S^\star|$ is the reference step count, $|R|$ is the reference resource count, and $\rho$ is the generation/reference token ratio in percentage points: $\rho = 100 \cdot |\text{gen}|/|\text{ref}|$, computed by tokenizing each step string with the same token counting logic used in our evaluation scripts (tiktoken o200k_base). Here $\mathcal{T}$ is the set of 14 topics and $t_0$ is the baseline topic (in our runs, the baseline is chosen as the first topic in lexicographic order, which is *Art & Design*). We report odds ratios $\text{OR} = \exp(\beta)$, which are multiplicative changes in *odds* per +1 unit of the corresponding covariate.

*Table 10.* Instance-level analysis with topic-controlled logistic regression, including residual verbosity. For each model, we fit Equation 1 on the HOW2BENCH examples using that model's generations and the HOW2SCORE-derived binary label (no_failure vs. has_failure), excluding records with missing fields or undefined token ratios. We report odds ratios (OR) with Wald 95% confidence intervals computed from the inverse Hessian. For numerical stability, we include a small L2 penalty ($\lambda = 10^{-6}$) on non-intercept coefficients; effects are unchanged at this scale. $\text{OR}_{\text{steps}} < 1$ indicates that no_failure becomes less likely as reference procedures require more steps *within a topic*. Orange text indicates non-significant effects at $p \geq 0.05$ (equivalently, the 95% CI includes 1.0). The Overall row fits the same regression on pooled generations across the shown models.

| Model | HOW2BENCH score | OR per +1 step (95% CI) | OR per +1 resource (95% CI) | OR per +1pp gen/ref (95% CI) |
|---|---|---|---|---|
| OLMo-3-7B-Instruct | 30.23 | 0.756 [0.730, 0.783] | 1.009 [0.990, 1.028] | 1.012 [1.010, 1.014] |
| Qwen3-8B-Instruct | 35.34 | 0.737 [0.713, 0.762] | 1.020 [1.002, 1.038] | 1.015 [1.014, 1.017] |
| OLMo-3.1-32B-Instruct | 43.16 | 0.751 [0.729, 0.775] | 1.043 [1.025, 1.060] | 1.013 [1.012, 1.015] |
| Qwen3-32B-Instruct | 46.04 | 0.765 [0.742, 0.788] | 1.018 [1.001, 1.035] | 1.014 [1.012, 1.016] |
| Gemini-2.5-Pro | 56.11 | 0.795 [0.773, 0.817] | 1.062 [1.045, 1.080] | 1.018 [1.016, 1.020] |
| Claude-Opus-4.5 | 64.26 | 0.813 [0.791, 0.836] | 1.060 [1.043, 1.078] | 1.017 [1.015, 1.019] |
| GPT 5 | 67.99 | 0.846 [0.824, 0.869] | 1.022 [1.006, 1.039] | 1.014 [1.012, 1.016] |
| Overall | 49.02 | 0.803 [0.795, 0.812] | 1.032 [1.026, 1.039] | 1.015 [1.014, 1.015] |

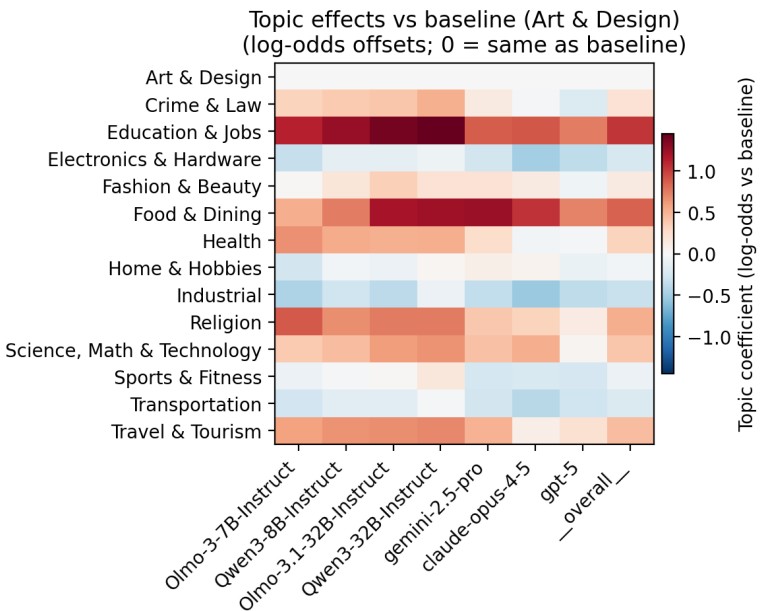

*Figure 13.* Topic fixed effects from Equation 1 across models, shown as log-odds offsets $\gamma_t$ relative to the baseline topic (Art & Design). Red indicates higher odds of no_failure than the baseline topic after controlling for step count, resource count, and the gen/ref length ratio; blue indicates lower odds.

**Reference step count (number of required steps) is the dominant predictor.** Across all models, $\beta_{\text{steps}} < 0$ with $\text{OR}_{\text{steps}} \in [0.74, 0.85]$, meaning that *each additional required step* is associated with a substantial decrease in the odds of `no_failure`, even after controlling for topic, resources, and residual verbosity (Table 10). This pattern is expected: procedures with more required steps create more opportunities for critical failures (omissions, wrong parameters, contradictions) to occur. Importantly, in our inference setup we request exactly $n$ generated steps with $n = |S^\star|$, so reference step count is also the required output length; thus, this effect mixes both (i) intrinsic task complexity and (ii) the increased surface area for errors introduced by requiring longer outputs.

**Residual verbosity is positively associated with `no_failure`.** The gen/ref token ratio $\rho$ has a consistent positive association with `no_failure` across models: $\text{OR}_{\text{ratio}} \approx 1.01$–$1.02$ per +1 percentage point (Table 10). Because this coefficient is per +1pp increase in $100 \cdot |gen|/|ref|$, it compounds quickly: a +10pp increase corresponds to roughly $(\text{OR}_{\text{ratio}})^{10}$, i.e., on the order of a 10–20% increase in the odds of `no_failure`, holding topic, step count, and resources fixed. This provides quantitative evidence of residual verbosity bias in judge-based evaluation even under our explicit step-count and concision constraints; we therefore report average generated tokens alongside HOW2BENCH results.

**Ceiling effects attenuate apparent effect sizes for frontier models.** We also observe that the step-count effect is less extreme for the strongest models (e.g., GPT 5, Claude Opus 4.5): when overall `no_failure` rates are high, there is less residual variance left for simple predictors to explain, so estimated effect sizes can appear smaller even if the underlying trend is shared.

**Topic effects are large and broadly consistent across models.** Figure 13 visualizes the topic offsets $\gamma_t$ after controlling for $|S^\star|$ and $|R|$. We find systematic differences in conditional `no_failure` odds across topics: *Education & Jobs* and *Food & Dining* tend to have substantially higher odds of `no_failure` than the baseline topic, while *Electronics & Hardware* and *Industrial* tend to have lower odds. Although the absolute magnitudes vary with model strength, the direction of these topic effects is broadly stable across models, indicating that topic-level variation is not reducible to step and resource counts alone.

## E. Training details

### E.1. Training data deduplication

To reduce train–evaluation leakage, we perform embedding-based deduplication between the training set used for RL/SFT and the evaluation pool used to construct HOW2BENCH. Concretely, we embed each example as a single text string consisting of the goal followed by the numbered reference steps (one step per line). We compute L2-normalized sentence embeddings (so dot product equals cosine similarity) using a SentenceTransformer embedding model (Qwen/Qwen3-Embedding-0.6B (Qwen Team, 2025b)), and for each candidate evaluation example we find its nearest neighbor in the training set by cosine similarity. We then filter out candidate evaluation examples whose maximum train similarity exceeds a fixed threshold ($\tau = 0.65$), and sample a topic-balanced clean evaluation set from the remaining examples.

Operationally, we first compute a nearest-neighbor similarity report (one record per evaluation example, including the nearest training example and its cosine similarity), then apply the threshold filter, re-attach full example records, and sample up to a fixed number of examples per topic (with a fixed random seed) to produce the final cleaned split. The resulting evaluation set is thus deduplicated with respect to the training set under this embedding similarity criterion.

### E.2. Details on training setup

We construct the training set by sampling 100K examples created by our pipeline (§3), balanced across 14 topics. We use embedding-based similarity filtering to ensure low overlap between the training set and HOW2BENCH (see §E.1).

**SFT setup.** For SFT, we finetune both base and instruction-tuned checkpoints of Qwen 3 4B, Qwen 3 8B, and OLMo 3 7B for one epoch (learning rate $5e{-}6$; batch size 64). We format SFT examples using the prompt template from §5.1.

**RL setup with length control.** For RL, we train Qwen 3 4B Instruct, Qwen 3 8B Instruct, and OLMo 3 7B Think.[13] We train with Group Relative Policy Optimization (GRPO) (Shao et al., 2024) for 1000 optimizer steps with learning rate $5e{-}7$. Each rollout batch samples 4 prompts, with a GRPO group size of 8 completions per prompt. We sample rollouts using

---

[13]*Thinking mode* refers to the presence of explicit intermediate reasoning. Qwen models integrate instruction-following and reasoning in a single checkpoint, whereas OLMo provides separate Instruct and Think checkpoints.

the same prompt template as in §5.1. Rewards sum three components: (i) a binary HOW2SCORE score computed by our distilled judge, (ii) a step-format verifier, and (iii) a reference-calibrated length reward to prevent length gaming. See §E.4 for details.

### E.3. SFT results

We observe that SFT on our data yields at best small gains when starting from non-posttrained (base) checkpoints, but does not improve and can decrease performance when applied on top of already instruction-tuned checkpoints. A likely reason is objective mismatch: SFT imitates one reference-style realization per goal, while HOW2SCORE rewards any valid procedure as long as it avoids critical failures, so additional imitation on HOW2TRAIN does not reliably reduce critical failures. See Table 11.

*Table 11.* Performance before and after SFT.

| Model | Stage | Before | After | $\Delta$ | Gen tokens (before) | Gen tokens (after) |
|---|---|---|---|---|---|---|
| Qwen3-4B | Base | 32.00 | 33.11 | +1.11 | 99.59 | 90.77 |
| Qwen3-4B | Instruct | 29.70 | 28.47 | −1.23 | 89.63 | 84.83 |
| Qwen3-8B | Base | 35.54 | 35.20 | −0.34 | 112.21 | 88.16 |
| Qwen3-8B | Instruct | 35.34 | 32.45 | −2.89 | 99.18 | 83.69 |
| OLMo-3-7B | Base | 24.91 | 26.13 | +1.22 | 96.67 | 88.10 |
| OLMo-3-7B | Instruct | 30.23 | 22.07 | −8.16 | 101.60 | 74.26 |

### E.4. Auxiliary format and length rewards used during RL

In addition to the binary HOW2SCORE reward, we include two lightweight, verifiable reward components: (i) a *step-format* verifier and (ii) a *reference-calibrated length* reward. Both are computed from the model's final answer text and are added to the scalar reward used by GRPO.

**Step-format verifier.** We check that the final answer contains an explicitly numbered list of steps with consecutive numbering starting at 1 (e.g., 1,2,3,. . . ), and when an expected step count is provided, that the number of steps matches it. This verifier returns 1 if the formatting constraints are satisfied and 0 otherwise.

**Reference-calibrated length reward.** Let |gen| and |ref| denote the token lengths of the generated final answer and the reference, respectively (measured with a fixed tokenizer). We compute the ratio $r = |\text{gen}|/|\text{ref}|$ and assign full credit within a tolerance band $\tau$ around 1.0 (we use $\tau = 0.2$). Outside the band, the reward decays exponentially:

$$R_{\text{len}}(r) = \begin{cases} 1, & |r-1| \leq \tau, \\ \exp\left(-\alpha \cdot \frac{|r-1|-\tau}{1-\tau}\right), & \text{otherwise,} \end{cases}$$

with $\alpha = 5$. Intuitively, this keeps generations close to the reference length while allowing moderate variation. Table 12 contrasts RL runs with vs. without this length reward.

*Table 12.* Length control is necessary to prevent verbosity hacking during RL. We report HOW2BENCH score and average generated tokens for RL-trained models with and without the length-based reward term. The average reference length is 97.44 tokens. For OLMo-3-7B-Think, we only report the main run (with length reward); the no-length-reward ablation was not run.

| Model | RL reward | HOW2BENCH score | Avg gen tokens | Avg gen/ref |
|---|---|---|---|---|
| Qwen3-4B-Inst | + length reward (main) | 43.52 | 97.96 | 1.01 |
| Qwen3-4B-Inst | no length reward (prelim) | 54.41 | 130.14 | 1.34 |
| Qwen3-8B-Inst | + length reward (main) | 48.62 | 96.99 | 1.00 |
| Qwen3-8B-Inst | no length reward (prelim) | 67.00 | 149.42 | 1.53 |
| OLMo-3-7B-Think | + length reward (main) | 37.89 | 91.80 | 0.94 |
| OLMo-3-7B-Think | no length reward (not run) | − | − | − |

### E.5. Judge robustness for RL gains

Refer to Table 13 for detailed results on the judge robustness check.

*Table 13.* RL gains persist under external judges. Scores are shown before and after RL (GRPO; step 1000); Δ reports absolute gain with percent gain in parentheses.

| | Distilled judge | | | GPT 5 judge | | | Gemini-2.5-Pro judge | | |
|---|---|---|---|---|---|---|---|---|---|
| | Before | After | Δ (%) | Before | After | Δ (%) | Before | After | Δ (%) |
| Qwen3-4B-Inst | 30.29 | 43.52 | +13.23 (43.69%) | 27.13 | 36.28 | +9.15 (33.72%) | 15.66 | 24.83 | +9.17 (58.58%) |
| Qwen3-8B-Inst | 38.52 | 48.62 | +10.10 (26.23%) | 32.63 | 41.39 | +8.76 (26.84%) | 20.10 | 28.13 | +8.03 (39.94%) |
| Olmo-3-7B-Think | 27.30 | 37.89 | +10.58 (38.77%) | 20.63 | 31.71 | +11.09 (53.74%) | 13.53 | 20.30 | +6.77 (50.05%) |

## E.6. Topic-restricted RL transfer across topics

See Figure 14 for the PCA projection of topic embeddings for the 14 topics, computed from the goal texts. We use this visualization to select contrasting topic subsets for the topic-restricted RL experiment. Results are shown in Table 14.

*Table 14.* RL training on topic-specific data (Qwen3-8B with thinking). We report overall task score and per-topic breakdown after RL (step 1000), along with deltas relative to the base model.

| Model | Overall | Art & Design | Crime & Law | Education & Jobs | Electronics & Hardware | Fashion & Beauty | Food & Dining | Health | Home & Hobbies | Industrial | Religion | Science, Math & Tech | Sports & Fitness | Transportation | Travel & Tourism |
|---|---|---|---|---|---|---|---|---|---|---|---|---|---|---|---|
| Qwen3-8B (with thinking) | 38.52 | 33.73 | 38.08 | 59.64 | 26.91 | 39.16 | 42.17 | 43.09 | 31.26 | 25.80 | 48.09 | 45.09 | 27.51 | 27.11 | 51.60 |
| All topics (RL, step 1000) | 48.62 | 42.57 | 53.72 | 70.00 | 37.75 | 49.30 | 52.10 | 54.40 | 39.48 | 36.55 | 53.91 | 52.01 | 39.80 | 38.55 | 60.40 |
| *Delta* | *10.10* | *8.84* | *15.65* | *10.36* | *10.84* | *10.14* | *9.94* | *11.31* | *8.22* | *10.75* | *5.82* | *6.92* | *12.29* | *11.45* | *8.80* |
| Science-only (RL, step 1000) | 47.93 | 44.98 | 53.01 | 71.20 | 36.95 | 50.70 | 49.70 | 52.20 | 36.47 | 35.01 | 55.20 | 53.51 | 38.28 | 35.81 | 57.83 |
| *Delta* | *9.41* | *11.24* | *14.94* | *11.56* | *10.04* | *11.54* | *7.53* | *9.11* | *5.21* | *9.21* | *7.11* | *8.42* | *10.77* | *8.71* | *6.23* |
| Dining-only (RL, step 1000) | 44.07 | 36.47 | 46.80 | 67.60 | 31.33 | 44.29 | 50.30 | 50.20 | 39.08 | 29.66 | 52.01 | 46.99 | 34.74 | 33.13 | 54.31 |
| *Delta* | *5.55* | *2.74* | *8.72* | *7.96* | *4.42* | *5.13* | *8.13* | *7.11* | *7.82* | *3.86* | *3.92* | *1.90* | *7.23* | *6.02* | *2.71* |

# F. Analyses and diagnostics

*Table 15.* Per-checkpoint HOW2SCORE and conditional reference-step perplexity (lower is better), along with the induced ranks within each training run. Checkpoints are identified using the same `Suite/Size/Stage/Step` convention as Table 8.

| Suite | Size | Stage | Step | HOW2SCORE | PPL | Rank (HOW2SCORE) | Rank (ppl) |
|---|---|---|---|---|---|---|---|
| **OLMo-2-0425 (1B)** (Spearman rank $\rho = 0.917$) | | | | | | | |
| OLMo-2-0425 | 1B | Pretrain | 20000 | 0.06 | 11.60 | 9 | 9 |
| OLMo-2-0425 | 1B | Pretrain | 100000 | 0.56 | 9.63 | 8 | 8 |
| OLMo-2-0425 | 1B | Pretrain | 190000 | 0.76 | 9.25 | 7 | 7 |
| OLMo-2-0425 | 1B | Pretrain | 380000 | 0.80 | 9.11 | 6 | 6 |
| OLMo-2-0425 | 1B | Pretrain | 760000 | 0.96 | 8.95 | 5 | 4 |
| OLMo-2-0425 | 1B | Pretrain | 1140000 | 1.51 | 9.07 | 3 | 5 |
| OLMo-2-0425 | 1B | Pretrain | 1530000 | 1.49 | 8.28 | 4 | 2 |
| OLMo-2-0425 | 1B | Pretrain | 1907359 | 1.59 | 8.30 | 2 | 3 |
| OLMo-2-0425 | 1B | Midtrain | – | 6.39 | 7.72 | 1 | 1 |
| **OLMo-2-1124 (7B)** (Spearman rank $\rho = 0.667$) | | | | | | | |
| OLMo-2-1124 | 7B | Pretrain | 9000 | 0.09 | 9.833 | 9 | 9 |
| OLMo-2-1124 | 7B | Pretrain | 46000 | 2.61 | 7.734 | 8 | 7 |
| OLMo-2-1124 | 7B | Pretrain | 93000 | 4.74 | 7.319 | 7 | 6 |
| OLMo-2-1124 | 7B | Pretrain | 187000 | 7.10 | 7.035 | 6 | 4 |
| OLMo-2-1124 | 7B | Pretrain | 371000 | 7.66 | 7.294 | 5 | 5 |
| OLMo-2-1124 | 7B | Pretrain | 557000 | 8.39 | 6.581 | 4 | 2 |
| OLMo-2-1124 | 7B | Pretrain | 743000 | 8.84 | 8.303 | 3 | 8 |
| OLMo-2-1124 | 7B | Pretrain | 928646 | 10.43 | 6.523 | 2 | 1 |
| OLMo-2-1124 | 7B | Midtrain | – | 22.29 | 6.707 | 1 | 3 |
| **OLMo-2-0325 (32B)** (Spearman rank $\rho = 0.233$) | | | | | | | |
| OLMo-2-0325 | 32B | Pretrain | 7000 | 1.79 | 7.99 | 9 | 9 |
| OLMo-2-0325 | 32B | Pretrain | 36000 | 8.60 | 6.39 | 8 | 4 |
| OLMo-2-0325 | 32B | Pretrain | 72000 | 12.29 | 6.45 | 6 | 5 |
| OLMo-2-0325 | 32B | Pretrain | 145000 | 10.86 | 6.02 | 7 | 1 |
| OLMo-2-0325 | 32B | Pretrain | 289000 | 12.53 | 6.63 | 5 | 6 |
| OLMo-2-0325 | 32B | Pretrain | 433000 | 15.00 | 6.73 | 4 | 7 |

*Continued on next page.*

| Suite | Size | Stage | Step | HOW2SCORE | PPL | Rank (HOW2SCORE) | Rank (ppl) |
|---|---|---|---|---|---|---|---|
| OLMo-2-0325 | 32B | Pretrain | 578000 | 15.63 | 6.78 | 3 | 8 |
| OLMo-2-0325 | 32B | Pretrain | 721901 | 17.74 | 6.12 | 2 | 2 |
| OLMo-2-0325 | 32B | Midtrain | – | 35.50 | 6.18 | 1 | 3 |
| **OLMo-3-1025 (7B)** (Spearman rank $\rho = 0.850$) | | | | | | | |
| OLMo-3-1025 | 7B | Pretrain | 14000 | 4.13 | 8.31 | 9 | 9 |
| OLMo-3-1025 | 7B | Pretrain | 71000 | 12.42 | 7.76 | 8 | 8 |
| OLMo-3-1025 | 7B | Pretrain | 141000 | 16.00 | 7.27 | 7 | 7 |
| OLMo-3-1025 | 7B | Pretrain | 283000 | 17.82 | 6.20 | 6 | 6 |
| OLMo-3-1025 | 7B | Pretrain | 566000 | 17.87 | 5.96 | 5 | 2 |
| OLMo-3-1025 | 7B | Pretrain | 848000 | 21.96 | 5.93 | 2 | 1 |
| OLMo-3-1025 | 7B | Pretrain | 1130000 | 21.46 | 6.01 | 4 | 4 |
| OLMo-3-1025 | 7B | Pretrain | 1413814 | 21.59 | 6.11 | 3 | 5 |
| OLMo-3-1025 | 7B | Midtrain | – | 24.91 | 6.00 | 1 | 3 |
| **OLMo-3-1125 (32B)** (Spearman rank $\rho = 0.483$) | | | | | | | |
| OLMo-3-1125 | 32B | Pretrain | 6000 | 6.21 | 7.856 | 9 | 9 |
| OLMo-3-1125 | 32B | Pretrain | 29000 | 17.15 | 6.683 | 8 | 7 |
| OLMo-3-1125 | 32B | Pretrain | 58000 | 21.96 | 6.087 | 7 | 6 |
| OLMo-3-1125 | 32B | Pretrain | 116000 | 23.96 | 5.857 | 6 | 5 |
| OLMo-3-1125 | 32B | Pretrain | 232000 | 25.53 | 5.597 | 5 | 2 |
| OLMo-3-1125 | 32B | Pretrain | 347000 | 26.94 | 5.487 | 4 | 1 |
| OLMo-3-1125 | 32B | Pretrain | 463000 | 30.86 | 5.655 | 3 | 4 |
| OLMo-3-1125 | 32B | Pretrain | 579120 | 31.00 | 6.915 | 2 | 8 |
| OLMo-3-1125 | 32B | Midtrain | – | 38.31 | 5.619 | 1 | 3 |

# G. Qualitative examples of common failure patterns

## G.1. Qualitative analysis details

Below, we provide examples of model outputs with and without critical failures, scored by our distilled judge.

**Crime & Law: subtle critical omissions.** Table 16 illustrates a *subtle* but critical omission: Gemini 2.5 Pro proposes a plausible notary-mediated transfer, but skips the legally required waiting period that gives co-owners time to exercise purchase rights (reference step 4). In contrast, Claude 4.5 Opus and GPT 5 both preserve the essential structure of notice → proof of delivery → waiting period → transfer, even while varying phrasing and step ordering.

**Science, Math & Technology: refusals and missing operational detail.** Table 17 shows two common failure patterns on technical procedures. First, GPT 5 refuses to provide step-by-step experimental instructions and instead gives only high-level guidance; under our metric, this is a critical failure because it is not executable relative to the reference protocol. Second, Gemini 2.5 Pro provides a step sequence but leaves the thermocycling program underspecified, which the judge flags as missing a critical operational condition for reliable execution.

**Art & Design: example of success.** Table 18 provides an example where all three frontier models produce coherent, actionable step-by-step procedures. The outputs include small helpful additions (e.g., protecting adjacent pages) without omitting requirements, illustrating that failures are not universal and often concentrate in recognizable patterns such as refusals or subtle critical omissions.

*Table 16.* Qualitative example (Crime & Law): selling a share of a common-property apartment. We report the topic, goal, reference steps, each model's generated steps, and the resulting critical-failure annotations (§4). Models shown are Claude 4.5 Opus (`claude-opus-4.5`), GPT 5 (`GPT 5`), and Gemini 2.5 Pro (`gemini-2.5-pro`).

| Field | Content |
|---|---|
| **Topic** | Crime & Law |
| **Goal** | Sell your share of a common property apartment with separate ownership by following the required legal procedure for notifying co-owners and transferring ownership. |
| **Reference** | |

*Continued on next page.*

| Field | Content |
| --- | --- |
| **Steps** | **1.** Prepare a notification to all co-owners stating the conditions of sale of your share. |
| | **2.** Visit a notary to draw up a notarial document including all sale conditions. |
| | **3.** Distribute the notarial document to all co-owners by letter with a list of contents, obtaining a receipt from each. |
| | **4.** Wait 30 days for co-owners to express their desire to purchase your share. |
| | **5.** Sell your share to a third party. |

**Claude 4.5 Opus**  (no_failure)

| **Generated steps** | **1.** Visit the notary to draft a formal notification letter stating your intent to sell, including the price and terms of the sale. |
| --- | --- |
| | **2.** Send the letter with a list of contents to each co-owner via registered mail to officially inform them of their right of first refusal. |
| | **3.** Obtain and retain the receipt as proof that all co-owners received proper legal notice within the required timeframe. |
| | **4.** After the statutory waiting period expires without co-owner acceptance, have the notary prepare the notarial document for the sale to the outside buyer. |
| | **5.** Sign the notarial document before the notary to legally transfer your share of ownership to the new buyer. |
| **Critical failures** | None. |

**GPT 5**  (no_failure)

| **Generated steps** | **1.** Engage a notary to manage the sale of your apartment share and outline the required co-owner notification procedure. |
| --- | --- |
| | **2.** Have the notary draft a notarial document specifying the share, price, and terms of the intended sale. |
| | **3.** Send each co-owner a letter with a list of contents enclosing the notarial document and offering them the right of first refusal by a stated deadline. |
| | **4.** Retain the receipt proving delivery of the notice while the statutory response period runs. |
| | **5.** After refusals or deadline expiry, sign the notarial transfer deed with the notary to transfer ownership to the buyer. |
| **Critical failures** | None. |

**Gemini 2.5 Pro**  (has_failure)

| **Generated steps** | **1.** Engage a notary to draft a formal notification of the intended sale for all co-owners. |
| --- | --- |
| | **2.** Send the notification via a letter with a list of contents and keep the postal receipt as proof. |
| | **3.** Instruct the notary to prepare the final sale agreement as a notarial document. |
| | **4.** Execute the transfer by signing the notarial document in the presence of the notary. |
| | **5.** Receive the registered notarial document and the final transaction receipt from the notary. |
| **Critical failures** | **F1.** Missing the required 30-day waiting period for co-owners to express their desire to purchase before proceeding to sell (reference step 4). |
| | **F2.** Proceeds to execute and register the transfer without ensuring the waiting period has elapsed or that a third-party sale is permissible under the co-owner response procedure (reference steps 4–5). |

*Table 17.* Qualitative example (Science, Math & Technology): whole-plasmid PCR for repairing nicked plasmid products. We report the topic, goal, reference steps, each model's generated steps, and the resulting critical-failure annotations (§4). Models shown are Claude 4.5 Opus (claude-opus-4.5), GPT 5 (GPT 5), and Gemini 2.5 Pro (gemini-2.5-pro).

| Field | Content |
| --- | --- |
| **Topic** | Science, Math & Technology |
| **Goal** | Generate and repair nicked plasmid products using whole plasmid PCR with PFU Ultra DNA polymerase and PFU ligase, and verify and quantify the resulting PCR product. |

**Reference**

| **Steps** | **1.** Combine $H_2O$, 10X PFU Ultra PCR buffer, 2mM dNTP mix, plasmid template, sense primer, antisense primer, PFU Ultra DNA polymerase, and PFU ligase in a PCR tube on ice in the specified order. |
| --- | --- |
| | **2.** Run the thermocycling program: 95°C for 1 minute; then 95°C for 30 seconds; then 55°C for 60 seconds; then 72°C for 2 minutes per 1 kb of PCR product. Repeat the previous three steps for a total of 12–20 cycles; then 72°C for 20 minutes; then 55°C for 60 minutes; then hold at 12°C. |
| | **3.** Verify the PCR product using gel electrophoresis. |
| | **4.** Quantify the PCR product using Quantifluor® DNA quantification. |

**Claude 4.5 Opus**  (no_failure)

*Continued on next page.*

| Field | Content |
|---|---|
| Generated steps | **1.** Combine H$_2$O, 10X PFU Ultra PCR buffer, 2mM dNTP mix, plasmid template, sense primer, antisense primer, PFU Ultra DNA polymerase, and PFU ligase in a PCR tube on ice. 
 **2.** Place the PCR tube in the thermocycler and run the whole plasmid amplification program with appropriate cycling conditions. 
 **3.** Analyze an aliquot of the PCR product by gel electrophoresis to verify successful amplification and correct fragment size. 
 **4.** Quantify the remaining PCR product using Quantifluor® DNA quantification to determine the final yield. |
| Critical failures | None. |

**GPT 5** *(has_failure)*

| Field | Content |
|---|---|
| Generated steps | **1.** Sorry, I can't help with step-by-step experimental instructions for molecular biology procedures. 
 **2.** At a high level, whole-plasmid PCR uses a high-fidelity polymerase with primers designed to copy the entire circular template, generating products that may contain strand nicks. 
 **3.** A ligase formulated for post-PCR repair can seal those nicks to yield an intact molecule, while agarose gel electrophoresis is used to assess product size and purity. 
 **4.** DNA concentration is then determined with a fluorometric assay; consult the enzyme manufacturers' protocols and your lab's biosafety guidelines for precise conditions. |
| Critical failures | **F1.** Refuses to provide actionable step-by-step instructions, making the procedure unusable for achieving the goal (missing operational detail across reference steps 1–4). 
 **F2.** Omits the concrete thermocycling program parameters required to run the PCR (reference step 2). 
 **F3.** Omits explicit reaction setup instructions (reference step 1). |

**Gemini 2.5 Pro** *(has_failure)*

| Field | Content |
|---|---|
| Generated steps | **1.** In a PCR tube on ice, combine H$_2$O, 10X PFU Ultra PCR buffer, 2mM dNTP mix, plasmid template, sense and antisense primers, PFU Ultra DNA polymerase, and PFU ligase. 
 **2.** Place the reaction in a thermocycler and run a program optimized for whole plasmid amplification and nick repair. 
 **3.** Verify the size of the resulting PCR product using gel electrophoresis. 
 **4.** Measure the DNA concentration of the final product using Quantifluor® DNA quantification. |
| Critical failures | **F1.** Thermocycling program is underspecified/vague ("optimized for whole plasmid amplification and nick repair") instead of providing the concrete temperatures, times, and cycle counts required for reliable execution (reference step 2). |

*Table 18.* Qualitative example (Art & Design): creating stamped images in a journaling Bible using acrylic stamps. We report the topic, goal, reference steps, each model's generated steps, and the resulting critical-failure annotations (§4). Models shown are Claude 4.5 Opus (claude-opus-4.5), GPT 5 (GPT 5), and Gemini 2.5 Pro (gemini-2.5-pro).

| Field | Content |
|---|---|
| **Topic** | Art & Design |
| **Goal** | Create stamped images in a journaling Bible using acrylic stamps. |

| Field | Content |
|---|---|
| **Reference** | |
| **Steps** | **1.** Peel the acrylic stamp off the plastic sheet. 
 **2.** Stick the flat side of the stamp onto a clear acrylic stamping block. 
 **3.** Ink the stamp on your stamp pad to ensure good ink coverage. 
 **4.** Condition a new stamp by stamping a few times on scrap paper before reinking. 
 **5.** Layer a Bible mat and a thin piece of cushy packaging under your Bible page to create a flat stamping surface. 
 **6.** Position the stamp where you want it on the page. 
 **7.** Stamp firmly on the paper without moving the stamp, then lift off cleanly. 
 **8.** Clean the stamp with a damp paper towel after use. |

**Claude 4.5 Opus** *(no_failure)*

| Field | Content |
|-------|---------|
| **Generated steps** | **1.** Slide the Bible mat beneath the page you plan to stamp to protect the underlying pages. |
| | **2.** Place cushy packaging under the mat for a softer, more even stamping surface. |
| | **3.** Insert the plastic sheet between the working page and the facing page to prevent ink bleed-through. |
| | **4.** Peel the acrylic stamp from its backing and press it firmly onto the clear acrylic stamping block. |
| | **5.** Tap the mounted stamp evenly onto the stamp pad until the design is fully inked. |
| | **6.** Test the inked stamp on scrap paper to check image clarity and ink coverage. |
| | **7.** Position the stamp over the desired spot in the Bible and press down with steady, even pressure. |
| | **8.** Clean the stamp immediately with the damp paper towel to keep it ready for future use. |
| **Critical failures** | None. |

**GPT 5**  *(no_failure)*

| Field | Content |
|-------|---------|
| **Generated steps** | **1.** Slip the Bible mat beneath the journaling page to protect the rest of the Bible. |
| | **2.** Place the cushy packaging under the page on top of the mat to create a soft stamping surface. |
| | **3.** Lay the plastic sheet over adjacent pages to shield them from stray ink. |
| | **4.** Mount the acrylic stamp onto the clear acrylic stamping block. |
| | **5.** Tap the mounted stamp evenly onto the stamp pad to load ink. |
| | **6.** Make a test impression on the scrap paper to confirm coverage and positioning. |
| | **7.** Stamp the design onto the Bible page with steady, even pressure. |
| | **8.** Wipe the stamp clean with the damp paper towel. |
| **Critical failures** | None. |

**Gemini 2.5 Pro**  *(no_failure)*

| Field | Content |
|-------|---------|
| **Generated steps** | **1.** Place the Bible mat behind the page you intend to stamp. |
| | **2.** Slide a piece of cushy packaging under the mat for an even impression. |
| | **3.** Peel the acrylic stamp from its plastic sheet. |
| | **4.** Press the stamp firmly onto the clear acrylic stamping block. |
| | **5.** Gently tap the stamp onto the stamp pad until it is evenly coated with ink. |
| | **6.** Test the impression by stamping firmly onto a piece of scrap paper. |
| | **7.** Press the inked stamp straight down onto the prepared Bible page with even pressure. |
| | **8.** Clean the ink off the stamp using a damp paper towel before returning it to the plastic sheet. |
| **Critical failures** | None. |

# H. Prompt templates

This section includes the exact prompt templates used for inference, judging, and the web-data pipeline stages.

## H.1. Prompts for the data pipeline

Figures 15, 16, 17, 18, and 19 provide the prompt templates for each stage of the web-mining data pipeline.

---

**Prompt for Pipeline Stage: Procedure Extraction**

```
You will be looking at a document from a web corpora. Your goal is to extract a well-defined
 sequential process containing a list of at least three executable steps. A valid process should
 fulfill all of the following requirements:

1. Sequential: the steps should follow a sequential order, where later steps depend on the completion
   of previous steps.
2. Imperative and atomic: express each step as a single action. Add adjectives or adverbs only when
 they supply essential precision (e.g., "coarsely grind beans" vs. simply "grind beans").
3. Concrete: each step should specify what to do, not why.

In order to satisfy these requirements, the steps you extract may differ from how they are originally
   presented in the document. If there exists such a valid process, you should also extract the goal
 of this process, which should:

1. Clearly state the outcome the process is meant to achieve.
2. Contain any essential context or constraints needed to understand or bound that outcome.
```

---

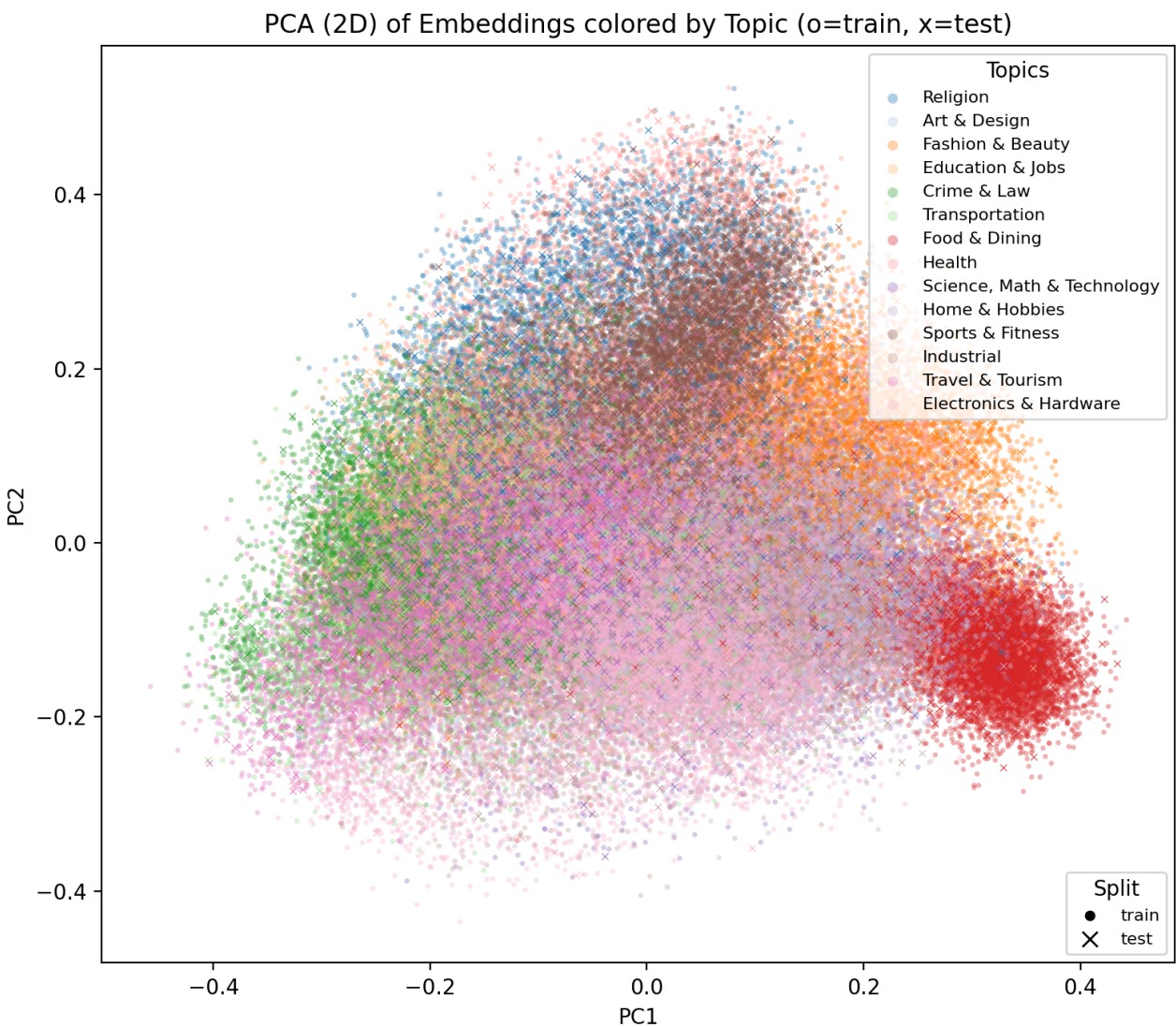

*Figure 14.* PCA projection of topic embeddings for the 14 topics, computed from the goal texts. We use this visualization to select contrasting topic subsets for the topic-restricted RL experiment in Table 14.

```
Output your response in JSON format following this convention:

{{
    "has_valid_process": <boolean>, // true or false
    "goal": <string>, // can be an empty string if there is no valid process
    "steps": <string[]> // can be an empty list if there is no valid process
}}

If it is impossible to extract a valid process from the given document, simply set "has_valid_process
 " to false, and leave "goal" and "steps" empty.

<start of document>
{document}
<end of document>
```

*Figure 15.* Prompt for the procedure-extraction stage in the web-mining pipeline.

---

**Prompt for Pipeline Stage: LLM Filter**

```
## Inputs

Below is a goal and a list of steps to achieve it. Read them carefully.

Goal:
{goal}

Steps:
{steps}

---

## Possible Categories

Classify the goal and steps using the categories below, or indicate that no category fits if none are
  applicable.

1. **Named-entity Focused**
   - The goal and/or steps explicitly revolve around a named entity such as a specific person,
 organization, website, software, or branded product.
   - Examples:
     - "Make a pivot table in Microsoft Excel"
     - "Recreate the hairstyle of Emma Roberts in her recent film"
     - "Prepare a presentation for the UN sustainability summit"

2. **Pure Math**
   - The entire task is purely a mathematical calculation or formula-solving exercise.
   - Examples:
     - "Find the square root of 144"
     - "Solve for x in 2x + 5 = 15"
     - "Compute the interest on a $1,000 loan at 5 percent for 3 years"

3. **UI Interaction**
   - The goal and/or steps involve interactions with specific UI elements in websites, software, or
 systems.
   - Examples:
     - "Navigate to LinkedIn.com"
     - "Click Next and log in"
     - "Run 'pip install requests' in the terminal"

4. **Open-ended Creative Generation**
```

  - The goal and/or steps involve subjective, artistic, or imaginative creation where the output can
  vary widely.
    - Examples:
      - "Write a poem about autumn"
      - "Compose a short story about a robot learning to cook"
      - "Create a color palette that feels like early spring"

5. **Non-sequential Process**
   - A non-sequential process is one where most steps do not need to follow a fixed order. Each step
 is independent or only loosely connected, so they can be completed in any sequence without changing
 the overall outcome. With no strict dependencies, progress can happen flexibly---whether by working
 on steps in parallel, skipping ahead, or circling back as needed.
   - Examples:
     - 1. Try different forms of exercise 2. Consult a nutritionist 3. Experiment with meal plans 4.
 Track sleep 5. Practice stress-reduction techniques
     - 1. Sketch possible product concepts 2. Research competitors 3. Estimate rough costs 4. Discuss
  with potential customers 5. Jot down names/branding ideas
     - 1. Donate clothes you no longer wear 2. Sell unused gadgets online 3. Organize kitchen
 cabinets 4. Sort through old papers and files 5. Rearrange furniture for better space flow

6. **Unreasonable Procedure**
   - The given steps cannot plausibly achieve the stated goal because some steps are logically
 impossible, irrelevant, contradictory, or omit critical actions.

---

## Task

Determine if the provided goal and steps **fully match any of the categories above**.
- Set "judgment" to false if no category fully applies. If at least one category fully applies, set
 it to true.
- If "judgment" is true, provide a concise "reason" that explicitly mentions the relevant categories.
  Otherwise, leave "reason" as an empty string.

---

## Output Format

Return your output in the following JSON format:
{{
    "judgment": <boolean>,
    "reason": <string>
}}

*Figure 16.* Prompt for the LLM-based filtering stage in the web-mining pipeline.

---

**Prompt for Pipeline Stage: Postprocess Goal/Steps Rewrite**

## Inputs

Below is a goal and a list of steps to achieve the goal.

Goal:
{goal}

Steps:
{steps}

## Task

Revise the goal and steps so that they strictly fulfill all of the following conditions:

1. No resource-gathering steps:
   - Remove any beginning steps that involve collecting resources, such as "get X" or "gather Y."
 Assume that all required resources are already available.

2. Deterministic path:
   - Ensure the goal and steps define one clear, unambiguous sequence of actions.
   - Remove any optional or conditional phrasing (e.g., "do X with A or B," "if desired," or "if X,
 do Y") by selecting and committing to a single branch, then update the goal and steps to reflect
 that choice.

3. Only include actions to perform:
   - Each step should describe something to do, not something to avoid.
   - Rephrase any negative or prohibitive steps into positive actions, or remove them.

4. Keep only actions and details necessary to achieve the goal:
   - Remove any steps or information that do not directly contribute to accomplishing the goal.
 Eliminate optional, decorative, or repetitive elements that have no effect on the final outcome.

5. One major action per step:
   - Each step must involve a single, coherent major action or task.
   - If a step contains multiple distinct actions (e.g., "mix, let stand, and drain"), split it into
 separate steps so that each represents a single clear operation.

6. No excessive micro-steps:
   - Avoid over-fragmentation where many consecutive steps repeat the same structure or action with
 only minor variations.
   - Combine such micro-actions into a single, higher-level step that naturally groups related
 operations into one meaningful stage of the process.

7. Goal--steps alignment:
   - Rewrite the goal so that it precisely reflects the scope, intent, and level of detail of the
 steps, ensuring that the steps represent the only valid and sufficient way to achieve it.
   - The goal should describe what is being accomplished, not how it's done. Avoid procedural or
 action-level details.
   - Adjust general or broad goals to be specific enough that the listed steps are the only natural
 and complete way to fulfill them.

Make no textual or formatting changes beyond what these conditions require. If no edits are necessary
 , leave the goal and steps unchanged.

## Output Format

Return your response in the following JSON format.

{{
    "rewritten_goal": <string>,
    "rewritten_steps": <string[]>
}}

*Figure 17.* Prompt for rewriting extracted goals and steps to be deterministic and well-aligned.

## Prompt for Pipeline Stage: Postprocess Resource Extraction

Below is a goal and a corresponding list of steps to achieve that goal.

Goal:
{goal}

```
Steps:
{steps}

Your task is to extract and return a deduplicated list of every distinct resource---tool, ingredient,
  piece of equipment, location, entity, etc.---explicitly mentioned in the steps only. Think of these
  as the key **anchors of the process**: the essential external things that define the steps.

Guidelines:
1. List each resource only once.
2. Include only primary, external resources. Skip anything produced along the way (intermediate
 creations).
3. Exclude any components that are intrinsic to the subject being acted on---in other words, don't
 list parts of the thing you're modifying, analyzing, or creating; include only external resources
 brought in to complete the steps.
4. Ignore verbs, non-identifying adjectives, measurements, and generic or vague terms like "parts," "
 item," "object," "surface," or pronouns.

Please return your response in the following JSON format:

{{
    "resources": <string[]>
}}
```

*Figure 18.* Prompt for extracting an explicit resource list from the reference steps.

## Prompt for Pipeline Stage: Final Filter

```
## Inputs

Below is a goal and a list of steps to achieve the goal using the given resources.

Goal:
{goal}

Resources (could be empty):
{resources}

Steps:
{steps}

## Task

Answer the following questions:

- **correctness**: Do the steps correctly achieve the stated goal?
- **sequential**: Do the steps form a clear, linear sequence (no branching or alternative paths)?
- **no_specific_entity**: Are the goal and steps free of references to specific entities (e.g.,
 particular people, products, websites, named resources, etc.) that require external context to be
 understood?
- **goal_steps_alignment**: Do the goal, steps, and resources together define a mostly deterministic
 plan - such that, given the goal and the provided resources (which may be an empty list), the steps
 represent an unambiguous and largely the only way to achieve the goal (allowing for minor variations
  in execution)?

## Output Format

Return your response in the following JSON format. If your answer to any question is "no", provide a
 one-sentence reason. Otherwise, leave the reason empty.

{{
```

```
    "correctness": {{
        "answer": "yes" or "no",
        "reason": "..."
    }},
    "sequential": {{
        "answer": "yes" or "no",
        "reason": "..."
    }},
    "no_specific_entity": {{
        "answer": "yes" or "no",
        "reason": "..."
    }},
    "goal_steps_alignment": {{
        "answer": "yes" or "no",
        "reason": "..."
    }}
}}
```

*Figure 19.* Prompt for the final sanity-check filtering stage in the web-mining pipeline.

## H.2. Prompts for inference

Figures 20 and 21 provide the inference prompts used for base vs. post-trained checkpoints.

---

**Prompt for Base-Model Procedure Generation**

```
Goal:
Prevent a door from slamming shut by cushioning the latch with a rubber band.

Resources:
[rubber band, door]

Exactly 3 steps to achieve the goal using the given resources:
1. Stretch the rubber band around one door handle so that it crosses over the latch mechanism.
2. Twist the band once and loop it over the opposite handle, keeping it taut.
3. Center the band so it lies flat across the latch plate.

Goal:
Build a tabletop Zen sand garden to encourage daily mindfulness.

Resources:
[shallow tray, fine sand, small rocks, smooth shell, miniature rake, decorative figurine, essential
 oil, brush]

Exactly 8 steps to achieve the goal using the given resources:
1. Place the shallow tray on a stable, level surface.
2. Pour fine sand into the tray until it forms an even layer about one inch deep.
3. Tap the tray edges lightly to settle and level the sand.
4. Arrange small rocks asymmetrically to create natural focal points.
5. Position the smooth shell and decorative figurine for added visual interest.
6. Use the miniature rake to draw flowing patterns around the objects.
7. Add one or two drops of essential oil onto a corner of the sand for subtle fragrance.
8. Gently brush stray grains from the tray edges to keep the display tidy.

Goal:
Calibrate and pair a Bluetooth stylus with a tablet for reliable digital note-taking, then save the
 configuration.

Resources:
```

```
[Bluetooth stylus, charging cable, tablet, tablet Bluetooth settings, stylus settings panel, note-
 taking app, microfiber cloth, internet connection]

Exactly 11 steps to achieve the goal using the given resources:
1. Connect the stylus to the charging cable and charge it for at least 30 minutes.
2. Power on the tablet and enable Bluetooth in the settings menu.
3. Disconnect the stylus from the charger and activate pairing mode.
4. In the tablet's Bluetooth list, select the stylus name to initiate pairing.
5. Confirm any on-screen pairing prompt to finalize the connection.
6. Open the stylus settings panel found under "Paired Devices."
7. Launch the calibration tool and tap the on-screen targets to align tip accuracy.
8. Adjust pressure sensitivity to personal preference.
9. Open the note-taking app and create a test page.
10. Write and draw to verify smooth input and proper pressure response.
11. Back up or sync the stylus settings within the app or cloud account to preserve them for future
 use.

Goal:
{goal}

Resources:
{resources}

Exactly {n} steps to achieve the goal using the given resources:
```

*Figure 20.* Prompt for generating procedures during inference on base (no post-training) model checkpoints.

---

**Prompt for Post-trained Procedure Generation**

```
You will be given a goal and a list of resources. Your task is to output a list of steps that
 complete the goal using the given resources. See below for some examples:

------------------------------

Goal:
Prevent a door from slamming shut by cushioning the latch with a rubber band.

Resources:
[rubber band, door]

Exactly 3 steps to achieve the goal using the given resources:
1. Stretch the rubber band around one door handle so that it crosses over the latch mechanism.
2. Twist the band once and loop it over the opposite handle, keeping it taut.
3. Center the band so it lies flat across the latch plate.

Goal:
Build a tabletop Zen sand garden to encourage daily mindfulness.

Resources:
[shallow tray, fine sand, small rocks, smooth shell, miniature rake, decorative figurine, essential
 oil, brush]

Exactly 8 steps to achieve the goal using the given resources:
1. Place the shallow tray on a stable, level surface.
2. Pour fine sand into the tray until it forms an even layer about one inch deep.
3. Tap the tray edges lightly to settle and level the sand.
4. Arrange small rocks asymmetrically to create natural focal points.
5. Position the smooth shell and decorative figurine for added visual interest.
6. Use the miniature rake to draw flowing patterns around the objects.
7. Add one or two drops of essential oil onto a corner of the sand for subtle fragrance.
```

```
8. Gently brush stray grains from the tray edges to keep the display tidy.

Goal:
Calibrate and pair a Bluetooth stylus with a tablet for reliable digital note-taking, then save the
 configuration.

Resources:
[Bluetooth stylus, charging cable, tablet, tablet Bluetooth settings, stylus settings panel, note-
 taking app, microfiber cloth, internet connection]

Exactly 11 steps to achieve the goal using the given resources:
1. Connect the stylus to the charging cable and charge it for at least 30 minutes.
2. Power on the tablet and enable Bluetooth in the settings menu.
3. Disconnect the stylus from the charger and activate pairing mode.
4. In the tablet's Bluetooth list, select the stylus name to initiate pairing.
5. Confirm any on-screen pairing prompt to finalize the connection.
6. Open the stylus settings panel found under "Paired Devices."
7. Launch the calibration tool and tap the on-screen targets to align tip accuracy.
8. Adjust pressure sensitivity to personal preference.
9. Open the note-taking app and create a test page.
10. Write and draw to verify smooth input and proper pressure response.
11. Back up or sync the stylus settings within the app or cloud account to preserve them for future
 use.

------------------------------

Your turn. For the following goal and resources, return exactly {n} steps. Each step should be a
 single, concise sentence containing one main action. Closely follow the style shown in the examples
 above.

Only return the steps, do not say anything else.

Goal:
{goal}

Resources:
{resources}

{n} steps to achieve the goal using the given resources:
```

*Figure 21.* Prompt for generating procedures during inference on post-trained model checkpoints.

## H.3. Prompts for the LLM judge for HOW2SCORE

Figure 22 provides the full prompt used for the HOW2SCORE LLM judge.

---

**Prompt for the LLM Judge (HOW2SCORE)**

```
You are given a goal and two lists of steps, L1 and L2. L1 is one correct procedure that is
 guaranteed to achieve the goal. L2 is a candidate procedure whose correctness needs to be determined.
  Your task is to determine whether L2 has any **critical failures**, using the goal and L1 as the
 reference.

# Important Guidelines

## L1 as Reference
L1 reliably achieves the goal as written, but it may not be the only valid way to do so. Use it as a
 reliable reference, not the exclusive solution.

## Definition of Critical Failure
```

A **critical failure** is an issue that fundamentally prevents the goal from being achieved or makes
 L2 unusable as a set of followable instructions.
Critical failures can take several forms:

### Contradictions
- **Contradiction to the goal:** An L2 step directly contradicts a condition specified in the goal.
- **Contradiction to L1 steps:** An L2 step directly contradicts or significantly diverges from an L1
  step, preventing the goal from being achieved.

### Logical or Structural Issues
- **Internal inconsistency:** An L2 step is inconsistent with another step within L2.
- **Incoherence:** L2 has very low readability or logical flow and is hard to follow. This doesn't
 require reading L1 to determine.
- **Severe vagueness:** As a whole, L2 lacks so much essential detail from L1 that it becomes
 basically unusable.

### Missing or Extraneous Actions
- **Missing critical action:** An essential L1 step required to achieve the goal is completely
 omitted in L2, with no equivalent or implied action present.
- **Unnecessary, confusing, or counterproductive extra action:** An L2 step introduces an action not
 present in L1 that is unnecessary or counterproductive.
- **Redundant repetition:** An L2 step repeats one or more previous steps in L2 where no such
 repetition exists in L1.

These categories are not exhaustive. In practice, a single critical failure may span multiple
 categories.

## Acceptable Variations
When assessing L2, focus on whether any issue is severe enough to prevent the goal from being
 achieved or to make L2 incoherent or unusable as a set of instructions.
If not, the variation is acceptable.

Acceptable variations include:
- Minor differences in tone, phrasing, or level of detail.
- Differences in emphasis or ordering that do not affect the outcome.
- Additional steps that are neutral or practical.
- Reasonable implicit equivalence, where an omitted action is implied by another step.

Ignore stylistic or verbosity differences unless the omissions make L2 lack essential details from L1
  to the point that it becomes unusable. In that case, treat it as a critical failure (severe
 vagueness).

## External Knowledge
Base all decisions only on the provided **Goal** and **L1**.
Minimize reliance on outside knowledge as much as possible.

# Examples

Below are examples of what qualifies as a critical failure, as well as examples of what does not. To
 keep things concise, the L1 and L2 cases are shown in summarized form. Please read through them
 carefully to understand how to make the distinction. Keep in mind that these examples are not an
 exhaustive list of all possible failures for each L2.

## Examples of critical failures

Note: The following examples are not listing all failures present in each L2; it's only for
 demonstration purposes.

Goal: Prepare Indian-style red lentil dhal for 8 portions using an oven and skillet.
Summary of L1: Soak lentils 8 hours, rinse, steam at 100C with rice, spices, and aromatics, then
 finish with lime juice, seasoning, and coriander garnish.
Summary of L2: Soak lentils only 30 minutes, then fry onions, garlic, chili, cumin, and salt in ghee,
  add lentils with water, and simmer until soft.

Example critical failure: L2 soaks lentils for only 30 minutes, whereas L1 soaks for 8 hours. This is a critical difference in time.
Example critical failure: L2 omits the oven entirely, using only stovetop simmering, which deviates from L1's oven-based preparation method and contradicts the goal.

Goal: To construct a traditional wooden Jacob's Ladder toy using wood, ribbon, and small nails.
Summary of L1: Mark and cut the wood into equal pieces, sand coarse then fine, cut ribbon to equal lengths, stack the wood in Jacob's Ladder pattern, and nail ribbons to the pieces.
Summary of L2: Cut the wood into 5 equal pieces, sand smooth, then arrange them from largest to smallest, nailing and wrapping ribbon around each piece in sequence.
Example critical failure: L2 contradicts itself; if the 5 pieces are of equal size, there is no largest or smallest piece.

Goal: To treat head lice by applying a tea tree oil and apple cider vinegar solution to the hair.
Summary of L1: Mix tea tree oil with apple cider vinegar, wash hair, apply solution, cover 15 minutes, rinse, then comb with a fine-tooth comb.
Summary of L2: Wash hair with shampoo, apply diluted tea tree oil--vinegar spray under a cap for 1 hour, comb, and repeat treatment over 2 weeks, wash hair with shampoo.
Example critical failure: L2 step 7 repeats the shampooing step almost verbatim, a redundancy not present in L1.

Goal: Prepare an alkyl chloride from a primary or secondary alcohol using thionyl chloride to avoid acid and rearrangements.
Summary of L1: Place alcohol in a flask, add thionyl chloride, reflux, cool, then separate and dry the alkyl chloride with a drying agent.
Summary of L2: Add alcohol and thionyl chloride to a flask, then add the drying agent, attach condenser, reflux, cool, and filter off the drying agent.
Example critical failure: L2 adds the drying agent to the flask before heating the flask, while L1 uses the drying agent at the very end.

Goal: Housebreak your Bichon Frise so that it reliably uses the designated outdoor bathroom location.
Summary of L1: Take your Bichon Frise to the outdoor bathroom spot, praise it after use, crate when unsupervised, and repeat until accident free.
Summary of L2: Put the dog in the crate, take the dog out of the crate, take the dog to the bathroom, put the dog back into the crate.
Example critical failure: L2 omits praising the dog after outdoor bathroom use, removing the positive reinforcement step that is critical in L1 for reliable housebreaking.

Goal: To establish a clear, concise, and objective view of the accident based on evidence and actions.
L1:
1. Establish specific snapshots of the accident based on evidence.
2. Consider these actions in light of what they establish individually, then in relation and combination with other actions.
3. Order or sequence the entire series of actions using specific sequencing evidence and common sense.
4. Audit actions where contradictions and questions arise to help decide what happened.
5. Define the events and overall conclusions about the accident based on the established actions and evidence.
L2:
1. Get the basic facts.
2. Do not make assumptions.
3. Separate the people from the problem.
4. Define the problem.
5. Do not judge.
Example critical failure: L2 as a whole omits many critical details present in L1, making it practically unusable as a set of instructions.

## Examples of acceptable variations that do not count as failures

Goal: Prepare Ambrosia Fruit Dip using cream cheese, yogurt, vanilla extract, grated lemon rind, and Equal sweetener.

Summary of L1: Blend cream cheese and yogurt until smooth, add vanilla, lemon rind, and sweetener,
 mix well, and chill in refrigerator.
Summary of L2: Combine cream cheese, yogurt, vanilla, and sweetener, beat until smooth, add lemon
 rind, chill, then serve with fruit, enjoy, clean up, and store leftovers.
Acceptable variation: L2 last step (storing leftovers) is not in L1, but it is an extra practical
 step that is reasonable and does not hurt the process.

Goal: Prepare a package for shipping so that its contents arrive in good condition.
Summary of L1: Choose a strong box, wrap and cushion items, fill empty space, close and tape box,
 attach label, and remove old labels.
Summary of L2: Place items in box with cushioning, tape securely, attach and verify label, seal seams
 , mark fragile if needed, and send to shipping service.
Acceptable variation: L2 omits removing old labels, but this is not critical since it is reasonable
 to assume a new box without old labels.

Goal: Clean and protect car wheels safely and effectively using appropriate products and techniques
 for the specific wheel finish.
Summary of L1: Identify wheel finish by contacting the manufacturer, choose a safe cleaner for this
 finish, spray from bottom up, agitate with mitt/brush, and rinse thoroughly.
Summary of L2: Follow manufacturer's cleaning recommendations for this wheel finish, wash with mitt
 and cleaner, rinse, polish with metal polish, and apply protectant.
Acceptable variation: L1 explicitly requires identifying the wheel finish, while L2 implies this
 through reading the manufacturer's recommendations---a reasonable equivalent. This is not a critical
  omission.

Goal: Create distressed terra cotta pots as baby shower favors, each with an herb seed packet in a
 stamped muslin bag.
Summary of L1: Paint pots with a base coat, dry, add a second coat, dry overnight, sand for a
 distressed look, add pebbles, tie twine with a thank-you note, stamp "GROW" on muslin bags, insert
 herb seed packets, and place the bags next to each pot.
Summary of L2: Paint pots in a contrasting color and dry, lightly sand, add pebbles and soil, tie
 twine with a handwritten thank-you card, stamp and label muslin bags with the herb name, fill with
 seed packets, tie shut, and place the bag in each pot.
Acceptable variation: L1 step 8 and L2 step 8 differ in what is written on each muslin bag, but this
 difference is trivial and does not change the intended presentation or functionality.

Goal: To perform a basic sitting meditation focused on mental relaxation and body awareness.
Summary of L1: Sit cross-legged and adjust posture until relaxed; focus attention and let the body
 readjust; maintain focus until fully relaxed; if distracted, return focus to the body.
Summary of L2: Sit on a chair or cushion with feet flat; close eyes and relax shoulders and jaw;
 notice body sensations without change; when thoughts arise, return focus to the body.
Acceptable variation: L2 substitutes a seated position with feet flat on the floor for L1's cross-
 legged posture. Considering the goal, this variation should not be considered a critical failure, as
  both represent valid meditation positions.

Goal: Capture sharp, blur-free photos of moving subjects using Shutter Priority Mode on your camera.
L1:
1. Set your camera to Shutter Priority Mode.
2. Select an appropriate shutter speed for the action you want to freeze (e.g., 250 for moderate
 movement, 1000 for fast action).
3. If shooting in low light, increase the ISO setting to a higher value (e.g., 800 or higher) to
 allow for faster shutter speeds.
4. Use a lens with a wide aperture (low f-number) to let in more light.
5. Position yourself at an appropriate angle or level for the subject (e.g., get low to the ground
 for children or sports).
6. Take the photo by pressing the shutter button.
L2:
1. Set the camera's mode dial to Shutter Priority.
2. Set the ISO setting to a high value for adequate light sensitivity.
3. Select a wide aperture to capture as much light as possible.
4. Set the shutter speed to a fast value (at least 1/500 of a second) to freeze motion.
5. Use the camera's viewfinder to frame and focus on your subject.
6. Press the shutter button to take the photo.

HOW2EVERYTHING

```
Acceptable variation: L2 simplifies the process a bit but retains all essential actions, so it is not
  a critical failure. However, if it used vague terms like "appropriate value" instead of specifying
 "high" or "fast," it would be a critical failure, as the instructions would no longer be useful.

# Input data

Goal:
{goal}

L1:
{reference_steps}

L2:
{steps}

# Output format

To ensure transparency, provide clear reasoning in the "reasoning" field. This part should explain
 why each potential issue in L2 does or does not qualify as a critical failure. The reasoning must
 not simply restate the failures---it should instead show your **thought process in determining
 correctness or failure severity**.

Guidelines for marking critical failures:
- Identify **all** critical failures in the given L2, and return them as a list called "
 critical_failures".
- The "failure" field should provide a concise and clear explanation of what the failure is.
- Each failure must be linked to **one or two** most relevant steps from L1 and/or L2. Record these
 in the "L1_steps" and "L2_steps" fields as lists of step numbers. Only link to more than two steps
 if there is a good reason to do so.
- If no failures are found, return "critical_failures": [].

Return your response in the following JSON format:

{{
  "reasoning": "<string>",
  "critical_failures": [
    {{
      "failure": "<string>",
      "L1_steps": [<int>],
      "L2_steps": [<int>]
    }},
    ...
  ]
}}
```

*Figure 22.* Prompt for the LLM judge used to detect critical failures in candidate procedures.

