# OpenReview forum: "How2Everything: Mining the Web for How-to Procedures to Evaluate and Improve LLMs"
_ICML.cc/2026/Conference — ICML 2026 regular_

### Official Review · Reviewer_eWeC · 2026-03-10

**Soundness:** 3
**Presentation:** 3
**Significance:** 3
**Originality:** 3
**Overall Recommendation:** 5
**Confidence:** 4

**Summary:**

This paper introduces HOW2EVERYTHING, a scalable framework for evaluating and improving LLMs' ability to generate step-by-step how-to procedures. The framework consists of HOW2MINE (a pipeline that extracts 351K procedures from ~1M web pages across 14 topics), HOW2BENCH (a 7K-example evaluation benchmark), and HOW2SCORE (an LLM-judge-based metric that detects critical failures, distilled into an open 8B model with 80.5% human agreement). The authors demonstrate clear scaling trends across model sizes and training stages, and show that RL with HOW2SCORE as a reward improves procedure generation by >10 points without systematic regressions on standard benchmarks.

**Compliance With Llm Reviewing Policy:**

Affirmed.

**Final Justification:**

This paper presents a well-designed end-to-end framework (HOW2EVERYTHING) for evaluating and improving LLMs' procedural generation capabilities, with thorough robustness analyses and informative scaling results. My main concerns in the initial review centered on: (1) reference-anchored evaluation penalizing valid alternatives, (2) the unrealistic evaluation setup with fixed step counts, (3) incomplete RL ablation, and (4) lack of comparison with existing benchmarks.

The authors' rebuttal adequately addressed these points. In particular, the clarification that format compliance is near-saturated before RL (making it unlikely to drive the gains), the comparison table distinguishing HOW2BENCH from related benchmarks, and the manual quality analysis of 200 sampled web pages (96% credible sources, 84% reasonable procedures) were convincing. The remaining limitations — such as the absence of a post-training memorization ablation and the ecological validity gap — are acknowledged by the authors and do not undermine the core contributions.

I maintain my score of 5 (Accept).

**Key Questions For Authors:**

1. In §7.2, the memorization experiment only evaluates the midtrained models directly without subsequent post-training. Given that §7.1 (Figure 4) clearly demonstrates that RL substantially amplifies capabilities accumulated during pretraining, could the modest memorization gains observed at midtraining (+3.3 for 7B, +6.1 for 32B) be further amplified after post-training (SFT/RL)? An experiment that applies the same post-training recipe to the memorization-controlled checkpoints would more convincingly rule out memorization as a confound in the full training pipeline.

2. How sensitive are the HOW2BENCH rankings to the choice of step count n? If models were allowed to freely determine the number of steps, would the relative ordering of models change substantially?

3. Have you considered evaluating on a subset where the reference procedures were verified by domain experts rather than by GPT-4.1, to provide a stronger ground-truth baseline?

**Limitations:**

Partially. The Impact Statement covers societal risks (biased web data, safety-sensitive procedures) and mitigations. However, key technical limitations are not adequately discussed: (1) the circularity in the evaluation chain where LLMs are used for data filtering, reference validation, judging, and RL reward simultaneously; (2) the memorization analysis (§7.2) only examines midtraining without post-training, leaving open whether memorization effects are amplified through the full pipeline; and (3) the ecological validity gap from providing exact step counts and resources at inference is framed as a design choice rather than acknowledged as a fundamental limitation.

**Strengths And Weaknesses:**

Strengths

1. The paper presents a well-designed end-to-end framework that addresses a practically important and understudied problem — evaluating procedural validity at scale — with a clear pipeline from data mining to evaluation to training, each component well-motivated.

2. The robustness analyses are thorough and commendable: the authors systematically investigate potential confounds including judge self-preference bias (§D.2), format compliance (§7.1), and memorization (§7.2), and validate RL gains under external judges (§E.5), strengthening confidence in the reported results.

3. The scaling analysis across pretraining checkpoints (Figure 1b, Table 8) is a valuable contribution, showing that HOW2BENCH provides meaningful signal from early pretraining (~10²¹ FLOPs) through frontier models, a desirable property that many existing benchmarks lack.

Weaknesses

1. Reference-anchored evaluation. HOW2SCORE relies on a single reference procedure as anchor, inherently penalizing valid alternative solution paths. The moderate inter-annotator agreement (Krippendorff's α = 0.593) confirms that "critical failure" remains subjective in this open-world setting.

2. Unrealistic evaluation setup. Providing the model with the goal, resource list, and exact step count n at inference time rarely reflects real-world usage. HOW2BENCH thus measures constrained procedure completion rather than open-ended generation, limiting its ecological validity.

3. Incomplete RL ablation. The RL reward combines HOW2SCORE, a format verifier, and a length reward, but only the length reward is ablated (§E.4). The contributions of HOW2SCORE vs. the format verifier are not disentangled.

4. No comparison with existing benchmarks. Despite discussing related benchmarks (CaT-Bench, PARADISE, WikiHow-based datasets), the paper does not evaluate on any of them, making it hard to assess the incremental value of HOW2EVERYTHING.

---

> ### Author Rebuttal · Authors · 2026-03-31
>
> We sincerely thank the reviewer for recognizing our end-to-end framing, robustness analyses, and scaling results. Below, we address the reviewer's comments.
>
> # Weaknesses
>
> ## 1. Reference-anchored evaluation may penalize valid alternatives.
>
> We agree that this limitation is real. Because HOW2SCORE is reference-anchored, some valid alternative procedures may still be penalized. Our claim is therefore not that HOW2SCORE is a perfect oracle for open-ended procedural generation, but that after narrowing the task with rewritten goals, resource lists, and final validation, it is stable enough for relative comparison of models and as a practical RL signal. This interpretation is supported by several results: our distilled judge reaches 80.5% agreement with the human majority label, and model rankings are unchanged when we rescore the same generations with judges from different families (App. D.2). We will revise the paper to state this limitation more explicitly.
>
> ## 2. Providing the goal, resource list, and exact step count is unrealistic.
>
> We agree this is more controlled than typical real-world usage, but it is intentional: as explained in §5.1 and App. D.1, conditioning on resources and exact step count reduces degrees of freedom and improves comparability across outputs in a judged setting where verbosity and granularity are major confounds. We will make this tradeoff more explicit in the revision.
>
> ## 3. The RL ablation does not disentangle HOW2SCORE from the format verifier.
>
> The format verifier checks only simple structural constraints: consecutive numbering and matching the requested step count (App. E.4). Two existing results suggest it cannot explain the gains. First, these constraints are already near-saturated before RL: post-training models have near-0% step-count mismatch and essentially 0% duplicate steps (App. D.5 / Table 9). Second, SFT improves format compliance but hurts HOW2SCORE (Table 11), whereas RL improves HOW2SCORE. Together, this suggests the format verifier is not the main driver of the benchmark gains.
>
> ## 4. No comparison with existing benchmarks.
>
> We thank the reviewer for bringing up this point. The main reason we did not include direct score comparisons is that the mentioned benchmarks target different task formulations:
>
> | Benchmark | Task formulation | What it tests |
> | :---- | :---- | :---- |
> | HOW2BENCH (ours) | Generate a full procedure | End-to-end procedural validity via critical-failure evaluation |
> | CaT-Bench | Predict step-order / dependency relations in a plan | Local causal-temporal understanding of plan steps |
> | PARADISE | Infer warnings/tips from a goal without intermediate steps | Implicit planning knowledge from goals |
> | WikiHow-style generation | Generate a reference-like procedure | Similarity to a reference procedure |
>
> We agree that direct comparison would help readers better position the work. In the revision, we will include a comparison table.
>
> # Questions
>
> ## 1. Could memorization gains at midtraining be amplified after post-training?
>
> We agree this would be a useful ablation. The core question is whether the post-training gains reflect genuine procedural improvement. Several existing results support this interpretation: repeated document exposure in §7.2 sharply lowers document perplexity but yields only modest, non-monotonic HOW2SCORE gains (Table 3), and topic-restricted RL transfers beyond the training topic (App. E.6 / Table 14). These findings suggest that the post-training gains are not primarily driven by memorized procedures, although some contribution from that mechanism cannot be fully excluded. We will clarify this limitation in the revision.
>
> ## 2. How sensitive are HOW2BENCH rankings to the choice of step count n?
>
> As noted in our response to Weakness 2, fixing n is an intentional benchmarking choice. Empirically, App. D.7 / Table 10 shows that larger required step counts make examples harder across all tested models, suggesting that n acts as a common difficulty knob rather than distorting the ranking in a model-specific way. We will make this clearer in the revision.
>
> ## 3. Have you considered evaluating on a subset with expert-verified references?
>
> We agree that a subset with expert-verified references would provide a stronger ground-truth. The main tradeoff is scalability: expert verification is hard to apply at scale, so we rely on automated filtering and validation. As a sanity check, **we ran a new analysis**: we manually examined 200 randomly sampled web pages by reading the linked content, and found 191/200 (96%) came from credible sources. The extracted procedures were reasonable in 167/200 cases (84%), unclear in 28/200 (14%) when confident assessment required domain expertise, and unreasonable in only 5/200 (2.5%). This provides partial support for the quality of the references, while still leaving expert verification as an important direction for future work. We will include this analysis in the revision.

---

> > ### Author Rebuttal · Reviewer_eWeC · 2026-03-31
> >
> > I thank the authors for their detailed rebuttal. The responses adequately address most of my concerns. I maintain my overall recommendation of 5.

---

### Official Review · Reviewer_jeYV · 2026-03-11

**Soundness:** 3
**Presentation:** 3
**Significance:** 3
**Originality:** 3
**Overall Recommendation:** 5
**Confidence:** 3

**Summary:**

To address the goal of generating step-by-step procedural guides, the authors specifically design three sub-pipelines: HOW2MINE, HOW2BENCH, and HOW2SCORE, corresponding to data mining, curated benchmark construction, and an evaluation protocol that checks whether a generated procedure contains a critical failure, respectively. They also conduct comprehensive experiments across different training stages and model scales, covering a wide range of both open-source and closed-source models. The experimental evaluation is very thorough. Based on these experiments, they draw two key insights:

1. Training on procedures improves how-to generation with no out-of-domain regression.

2. Improvements in procedure generation are not driven by format compliance or memorization.

**Compliance With Llm Reviewing Policy:**

Affirmed.

**Key Questions For Authors:**

See weaknesses.

**Limitations:**

yes

**Strengths And Weaknesses:**

### Strengths
1. The paper is well motivated. “Generating step-by-step procedural guides” is indeed an important component for the practical deployment of current LLM agents, and therefore has strong research significance.

2. The experiments are thorough. The paper develops How2EVERYTHING across different stages and evaluates it under different training stages as well as across different open-source and closed-source models. The experimental setup is very comprehensive.

3. The writing is detailed. The paper itself is highly informative, with clear and detailed explanations of various concepts, making it easy to read and understand, while also being quite comprehensive.

### Weaknesses
1. In the HOW2SCORE pipeline, the idea of using an LLM-as-judge seems to rely directly on GPT-5 for evaluation. Although it demonstrates good alignment with human evaluation, is this approach perhaps too simplistic? Additionally, could there be an extra agent-style workflow to further improve this process?

2. The abstract and introduction do not seem to mention how HOW2BENCH is derived from HOW2MINE. In my view, this transition is important and should be mentioned in the corresponding sections.

3. Contributions 4 and 5 are למעשה experimental insights. It may be more appropriate to combine them into a single point, though this does not affect the quality of the paper itself and is merely a matter of writing style.

---

> ### Author Rebuttal · Authors · 2026-03-31
>
> We sincerely thank the reviewer for their time and for recognizing the paper’s motivation, comprehensiveness, and clarity. Below, we address the reviewer's comments.
>
> # Weaknesses
>
> ## 1. HOW2SCORE relies on GPT-5 / is the LLM-as-judge pipeline too simplistic? Could an agent workflow improve it?
>
> Our goal in HOW2SCORE is not to build the most elaborate verifier possible, but to build one that is scalable, efficient enough for large-scale evaluation and RL training, reproducible, and validated against humans. This emphasis on scalability is especially important for RL, where the judge must run cheaply and asynchronously over many rollouts. Agent-style verification would add substantial latency and make large-scale async rollouts much more cumbersome.
>
> Concretely:
> - We first compare several frontier judges against human labels on 200 examples and choose the strongest one (GPT-5) as the teacher because it has the best agreement (§4.3).
> - We then distill this judge into an open Qwen3-8B model using 73K GPT-5 annotations, yielding 90.5% agreement with GPT-5 and 80.5% agreement with the human majority label (§4.4).
> - Thus, GPT-5 is used as a teacher during judge development, while the large-scale evaluator used in the paper and in RL is the distilled Qwen3-8B judge.
> - This gives us a low-cost, reproducible judge for both evaluation and RL, without requiring proprietary APIs at test time.
>
> An agent-style verifier could be interesting future work, but we would view it as a different point in the cost/complexity tradeoff. Our emphasis is that a relatively simple but carefully defined binary critical-failure protocol already produces useful signals, tracks human judgments reasonably well, and scales to 7K-example evaluation plus RL.
>
> ## 2. The abstract/introduction do not clearly say how HOW2BENCH is derived from HOW2MINE.
>
> We thank the reviewer for pointing out the need to clarify the relationship between HOW2BENCH and HOW2MINE. The relationship is described in §3.2 ("From this pool, we construct HOW2BENCH by sampling 500 instances per topic (7,000 total), and reserve the rest as training data"), but this should be surfaced earlier. We will revise the abstract/introduction to make the pipeline explicit in one sentence, e.g., that HOW2MINE extracts 351K procedures and HOW2BENCH is built by sampling 500 instances per topic from this mined pool.
>
> ## 3. Contributions 4 and 5 are more like experimental insights.
>
> We agree with this comment. This is mostly a presentation issue, and we will merge these two points into one contribution in the revision.
>
> # Questions
> Same as weaknesses.

---

> > ### Author Rebuttal · Reviewer_jeYV · 2026-04-03
> >
> > Considering the quality of the paper itself, I choose to maintain my score.

---

### Official Review · Reviewer_A6qg · 2026-03-13

**Soundness:** 2
**Presentation:** 2
**Significance:** 2
**Originality:** 2
**Overall Recommendation:** 4
**Confidence:** 3

**Summary:**

This paper focus on measuring performance of procedure how to problems. The how2everything framework is composed with: 1) how2mine that extract and rewrites procedures from web corpuse, 2) how2bench a benchmark  with balanced topics to evaluate the LLM procedure reasoning, and 3) how2score -- a LLM judge based scoring to judge if a generation contain critical errors. The authors shows that How2Bench is able to measure models with different sizes and differentiate. In addition, they also show that RL with how2score helps the model improve on how2bench without significant out of domain influence.

**Compliance With Llm Reviewing Policy:**

Affirmed.

**Final Justification:**

Authors addressed most of the concerns in rebuttal

**Key Questions For Authors:**

see above

**Limitations:**

Yes

**Strengths And Weaknesses:**

The contributed HOW2MINE framework can be useful for the research community. However, it would be good to understand the procedure a bit more. It looks like all of the steps needs LLM apart from Heuristics filter. Seems quite a costly pipeline to run. Another concern is that the inter annotator agreement after iterations of training have Krippendorff’s α = 0.593, which is moderate but not strong. This makes the validity of the label a bit questionable. In addition, it also made later LLM based judge agreement with human label less meaningful. The RL gains looks good, I am wondering how robust are the RL gains under a purely human evaluation of post-RL outputs? Is the gain mainly due to the reward is also how2score?

---

> ### Author Rebuttal · Authors · 2026-03-31
>
> We sincerely thank the reviewer for their time and for recognizing the potential community value of HOW2MINE. Below, we address the reviewer's comments.
>
> # Weaknesses
>
> ## 1. The HOW2MINE pipeline seems costly and mostly LLM-based.
> We agree that cost matters. We report the end-to-end cost explicitly: mining 980K documents costs around $5,717 with the OpenAI batch API (§3.2).
>
> The pipeline is LLM-assisted by design because the target object, namely goal-conditioned and realistic procedures, is difficult to recover with heuristics alone. The multi-stage process progressively enforces structure and validity via extraction, heuristic filtering, LLM filtering, post-processing, and final validation (§3.2). This cost is amortized across 351K procedures from 189K domains spanning 14 topics (§3.1-§3.2), supporting both benchmark and training data construction.
>
> ## 2. Krippendorff’s α = 0.593 is moderate; does this weaken label validity?
> We agree the task is subjective, but our claim is narrower: the HOW2SCORE evaluation protocol, which asks whether a generated procedure contains any critical failure relative to the reference, is reliable enough for relative model comparison on HOW2BENCH and as a practical reward signal for RL (§4.3). This is an open-world, non-executable setting with many valid procedures, so disagreement often reflects ambiguity about whether a deviation is severe enough to cause failure.
>
> Several results support this interpretation: agreement improved substantially from the initial pilot (α=0.273) after codebook refinement and annotator screening (§4.2-§4.3); requiring agreement on the first failure location reduces agreement further (α=0.307), motivating our binary formulation (§4.3); GPT-5 reaches 83.0% agreement with the human majority label, while leave-one-out human agreement is 84.7%-88.5% (§4.3, Fig. 3); and our distilled 8B judge still achieves 80.5% agreement with the human majority label (§4.4). Together, these results suggest the labels are noisy but sufficiently stable for our intended use.
>
> ## 3. How robust are the RL gains under a purely human evaluation of post-RL outputs?
> We agree that direct human evaluation is the most relevant additional check. As a sanity check, **we ran a new blinded manual evaluation** on 200 randomly sampled HOW2BENCH examples comparing Qwen3-8B before vs. after 1000 RL steps. The evaluator for this analysis was one of the authors. To minimize bias, for each example, the evaluator saw only the goal, the reference steps, and two anonymous outputs labeled A and B; the mapping of A/B to pre-RL vs. post-RL was hidden and randomized separately for every example. Rather than selecting a preferred output, the evaluator labeled A and B independently for whether each contained a critical failure under the paper's rubric.
>
> In this blinded manual evaluation, the post-RL model improves from 32.5% no-failure (65/200) to 40.0% no-failure (80/200), a **+7.5 point gain**. In the paired breakdown, 23 examples are RL-only wins versus 8 base-only wins (McNemar exact p=0.0107). We view this as a sanity check showing that the RL gains are reflected in blinded manual evaluation, not only in the distilled HOW2SCORE judge. We will include this analysis in the revision.
>
> These new results complement existing observations in the manuscript: the RL gains persist under external judges not used in training. App. E.5 / Table 13 shows consistent improvements under GPT-5 (+9.15/+8.76/+11.09) and Gemini-2.5-Pro (+9.17/+8.03/+6.77) across all three models.
>
> # Questions
>
> Same as weaknesses.

---

> > ### Author Rebuttal · Reviewer_A6qg · 2026-04-03
> >
> > thanks for the reply, I have updated my score accordingly

---

### Decision · Program_Chairs · 2026-04-30

**Decision:**

Accept (regular)

**Comment:**

The authors introduce HOW2EVERYTHING, which is a framework for making and testing how-to guides with LLMs. The submission includes a mining pipeline (HOW2MINE) for grabbing hundreds of thousands of procedures, a balanced benchmark (HOW2BENCH), and an evaluation protocol (HOW2SCORE) that uses an LLM judge to find critical failures in steps. One of the main strengths here is how complete the work is. It doesnt just stop at data; it shows how you can use this signal for RL and actually improve model performance without breaking things on other standard benchmarks.

The reviews were generally quite positive, though some initial concerns were raised about the cost of the pipeline and the "LLM-as-a-judge" setup. I've looke through the authors' rebuttals and I am satisfied with the clarifications. Specificaly, the authors provided a new blinded manual evaluation that showed real gains from the RL, which helps prove that the improvements aren't just the model gaming the LLM judge. They also clarified that while the initial pipeline costs some money, the resulting 8B open-source judge makes the whole thing very accessible for the community to actually use.

There was also some back-and-forth about whether the task is a bit "artificial" because it provides the model with the exact number of steps and resources. While this is true, the authors argue this is a design choice to make comparisons more fair and reduce "verbosity bias," which makes sense in a benchmarking context. The scaling analysis across different training stages is also quite insightful.